# MemoryVLA: Perceptual-Cognitive Memory in Vision-Language-Action Models for Robotic Manipulation

**Hao Shi**[1†]    **Bin Xie**[2]    **Yingfei Liu**[2]    **Lin Sun**[4†]    **Fengrong Liu**[5†]    **Tiancai Wang**[2]
**Erjin Zhou**[2]    **Haoqiang Fan**[2]    **Xiangyu Zhang**[3,6]    **Gao Huang**[1✉]
[1]Department of Automation, BNRist, Tsinghua University  [2]Dexmal
[3]MEGVII Technology  [4]Tianjin University  [5]Harbin Institute of Technology  [6]StepFun
`shi-h23@mails.tsinghua.edu.cn`   `gaohuang@tsinghua.edu.cn`
[†] Work done during interning at Dexmal.    [✉] Corresponding author.

## Abstract

Temporal context is essential for robotic manipulation because such tasks are inherently non-Markovian, yet mainstream VLA models typically overlook it and struggle with long-horizon, temporally dependent tasks. Cognitive science suggests that humans rely on working memory to buffer short-lived representations for immediate control, while the hippocampal system preserves verbatim episodic details and semantic gist of past experience for long-term memory. Inspired by these mechanisms, we propose MemoryVLA, a Cognition-Memory-Action framework for long-horizon robotic manipulation. A pretrained VLM encodes the observation into perceptual and cognitive tokens that form working memory, while a Perceptual-Cognitive Memory Bank stores low-level details and high-level semantics consolidated from it. Working memory retrieves decision-relevant entries from the bank, adaptively fuses them with current tokens, and updates the bank by merging redundancies. Using these tokens, a memory-conditioned diffusion action expert yields temporally aware action sequences. We evaluate MemoryVLA on 150+ simulation and real-world tasks across three robots. On SimplerEnv-Bridge, Fractal, LIBERO-5 suites and Mikasa-Robo, it achieves 71.9%, 72.7%, 96.5%, and 41.2% success rates, respectively, all outperforming state-of-the-art baselines CogACT and $\pi_0$, with a notable +14.6 gain on Bridge and +11.8 gain on Mikasa-Robo. On 12 real-world tasks spanning general skills and long-horizon temporal dependencies, MemoryVLA achieves 84.0% success score, with long-horizon tasks showing a +26 improvement over state-of-the-art baseline. Project Page: https://shihao1895.github.io/MemoryVLA

## 1 Introduction

Vision-Language-Action (VLA) models (Brohan et al., 2023; Kim et al., 2024; Black et al., 2024; Li et al., 2024a; Sun et al., 2025; Xie et al., 2025), powered by large-scale cross-embodiment robotic datasets (O'Neill et al., 2024; Brohan et al., 2022; Khazatsky et al., 2024; Bu et al., 2025a) and pretrained Vision-Language Models (VLMs) (Karamcheti et al., 2024; Liu et al., 2023b; Bai et al., 2023a), have achieved remarkable progress in robotic manipulation. However, mainstream VLA models such as OpenVLA (Kim et al., 2024) and $\pi_0$ (Black et al., 2024) rely solely on the current observation, thereby overlooking temporal dependencies and performing poorly on long-horizon temporal manipulation tasks. As shown in Fig. 1 (a), *Push Buttons* tasks exhibit almost no visual difference before and after pushing, making it difficult to determine whether the action has already been completed. This highlights the non-Markovian nature of manipulation, where earlier actions influence later decisions, calling for temporal modeling. A naive strategy is to concatenate consecutive frames as input to the VLM. However, it faces two critical limitations: (1) The quadratic complexity of self-attention severely limits the usable temporal context length; (2) Sequential frame inputs are misaligned with the model's single-frame robotic pretraining distribution.

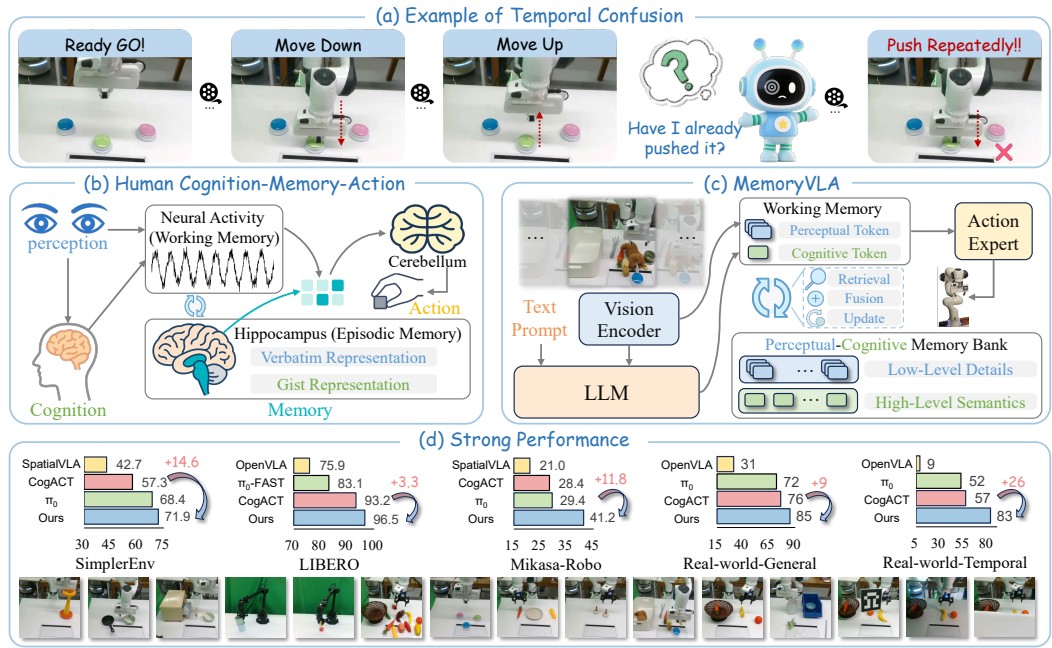

Figure 1: (a) In *Push Buttons* tasks, pre- and post-push states look nearly identical, calling for temporal modeling. (b) Humans handle manipulation tasks via a dual-memory system: working memory (neural activity) supports short-term control, while episodic memory (hippocampus) preserves long-term experience. (c) Inspired by this, MemoryVLA introduces a Perceptual–Cognitive Memory Bank that consolidates low-level perceptual details and high-level cognitive semantics for temporally aware decision making. (d) MemoryVLA outperforms state-of-the-art baselines.

Research in cognitive science (Baddeley & Hitch, 1974; Tulving et al., 1972; Reyna & Brainerd, 1995) demonstrates that humans handle manipulation tasks through a dual-memory system (Fig. 1 (b)). The brain encodes multi-modal sensory inputs into both perceptual and cognitive representations. These representations are buffered in working memory via transient neural activity, providing short-term retention for immediate decision-making. Concurrently, episodic memory, the long-term memory system supported by hippocampus, encodes past experiences with temporal index in two forms: verbatim representations preserving precise details and gist representations capturing abstract semantics. During execution, working memory retrieves decision-relevant contexts from episodic memory and integrates them with current representations to guide actions through cerebellar control, while simultaneously consolidating new experiences into episodic memory.

Drawing on cognitive science insights, we propose **MemoryVLA** (Fig. 1 (c)), a Cognition-Memory-Action framework for robotic manipulation that explicitly models temporal dependencies through a Perceptual–Cognitive Memory Bank (PCMB). First, a vision encoder extracts perceptual tokens from observation, while a large language model (LLM) processes them together with the language instruction, leveraging commonsense priors to produce cognitive tokens. Perceptual and cognitive tokens jointly form the working memory. Second, the PCMB stores both low-level perceptual details and high-level cognitive semantics over long horizons. During retrieval, working memory buffers current tokens and queries the PCMB with temporal positional encodings to fetch decision-relevant historical contexts, which are adaptively fused with current tokens via a gating mechanism while simultaneously updating the PCMB. When capacity is reached, temporally adjacent and semantically similar entries are consolidated to preserve essential information compactly. Finally, a memory-conditioned diffusion action expert is conditioned on cognitive tokens, with perceptual tokens enriching them with fine-grained details, to produce temporally aware robotic action sequences.

We conduct extensive evaluations of MemoryVLA across 3 robots and 150+ tasks with 500+ variations in simulation and real world. For SimplerEnv (Li et al., 2024b), MemoryVLA achieves 71.9% and 72.7% success rates on Bridge and Fractal suites, surpassing CogACT by 14.6 and 4.6 points, further outperforming $\pi_0$. For LIBERO (Liu et al., 2023a), it achieves 96.5% success rate across 5

suites (Spatial, Object, Goal, Long-10, and Long-90), exceeding both CogACT and $\pi_0$. For Mikasa-Robo (Cherepanov et al., 2025), it achieves 41.2% success rate, outperforming $\pi_0$ by 11.8 points. For real-world evaluations, we introduce 12 tasks across Franka and WidowX robots, spanning 6 general tasks and 6 long-horizon temporal tasks. MemoryVLA achieves 85% and 83% scores on general and temporal tasks, outperforming CogACT by 9 and 26 points, and substantially surpassing $\pi_0$. Moreover, MemoryVLA exhibits strong robustness and generalization under out-of-distribution conditions involving varied backgrounds, distractors, objects, containers, lighting and occlusion.

Our contributions are summarized as follows:

- Inspired by human memory systems from cognitive science, we propose MemoryVLA, a Cognition-Memory-Action framework that leverages VLM commonsense priors, a perceptual-cognitive memory mechanism, and a diffusion action expert to capture long-horizon temporal dependencies for robotic manipulation.
- We design a Perceptual–Cognitive Memory Bank with working memory that enables memory retrieval of decision-relevant contexts across high-level cognition and low-level perception, memory fusion that adaptively integrates them with current representations, and memory consolidation that merges temporally adjacent, semantically similar entries.
- MemoryVLA achieves state-of-the-art performance on SimplerEnv, LIBERO, Mikasa-Robo, and real-world. It also demonstrates strong robustness and generalization. On challenging long-horizon real-world tasks, it outperforms CogACT and $\pi_0$ by significant margins, underscoring the importance of temporal memory modeling.

## 2 RELATED WORKS

**Vision-Language-Action Models**    Driven by advances in visual foundation models (Radford et al., 2021; Caron et al., 2021; Liu et al., 2024a; Zheng et al., 2024a; 2025a; Zhang et al., 2025b; Wu et al., 2025; Wang et al., 2025), robot imitation learning has progressed rapidly yet remains confined to small, task-specific policies with limited generalization (Shridhar et al., 2023; Zhao et al., 2023; Chi et al., 2023; Goyal et al., 2023; Shi et al., 2025). To overcome these, the success of VLMs (Achiam et al., 2023; Touvron et al., 2023; Liu et al., 2023b; Bai et al., 2023b; Guo et al., 2025) and large-scale robot datasets (e.g., OXE (O'Neill et al., 2024), Agibot (Bu et al., 2025a)) spawned the vision-language-action (VLA) paradigm (Kim et al., 2024; Black et al., 2024; Yue et al., 2024; Zhong et al., 2025; Gao et al., 2025; Yang et al., 2025b). RT-2 (Zitkovich et al., 2023) and OpenVLA (Kim et al., 2024) tokenize continuous actions into discrete tokens and use VLMs for autoregressive prediction as if generating language. In contrast, $\pi_0$ (Black et al., 2024), CogACT (Li et al., 2024a), DexVLA (Wen et al., 2025) and HybridVLA (Liu et al., 2025c) adopt diffusion-based policies (Chi et al., 2023; Liu et al., 2024b) as action heads, leveraging iterative denoising to sample continuous control trajectories that capture diverse multimodal behaviors. However, none of these methods explicitly model temporal dependencies. Robotic manipulation is inherently non-Markovian, and neglecting history leads to failures on long-horizon temporal tasks.

**Temporal Modeling in Robotics**    Temporal modeling has been extensively studied in computer vision and autonomous driving (Wang et al., 2023; Liu et al., 2023c; Feng et al., 2023; Zhou et al., 2024), yet it has not been fully explored in robotic manipulation. Octo (Mees et al., 2024), RoboVLMs (Liu et al., 2025b), and Interleave-VLA (Fan et al., 2025) adapt the VLM paradigm to model robotic video data in an interleaved image-text format. While conceptually elegant, this format is complex to implement and computationally expensive, hindering its widespread application. RoboFlamingo (Li et al., 2023) compresses vision-language representation into a latent token and propagate it via LSTM (Hochreiter & Schmidhuber, 1997). The latent representation is obtained in a relatively coarse manner and the fine-grained perceptual history is largely discarded. TraceVLA (Zheng et al., 2024b) takes a different route, painting historical states as trajectories on the current frame, yet discards rich semantic details. UniVLA (Bu et al., 2025b) incorporates past actions into input prompts, making an initial attempt at temporal modeling. However, it merely serves as a Chain-of-Thought (Wei et al., 2022) process without effectively utilizing historical information. In contrast, we model both high-level cognitive semantics and fine-grained perceptual details within a memory framework, enabling effective temporal modeling for long-horizon manipulation.

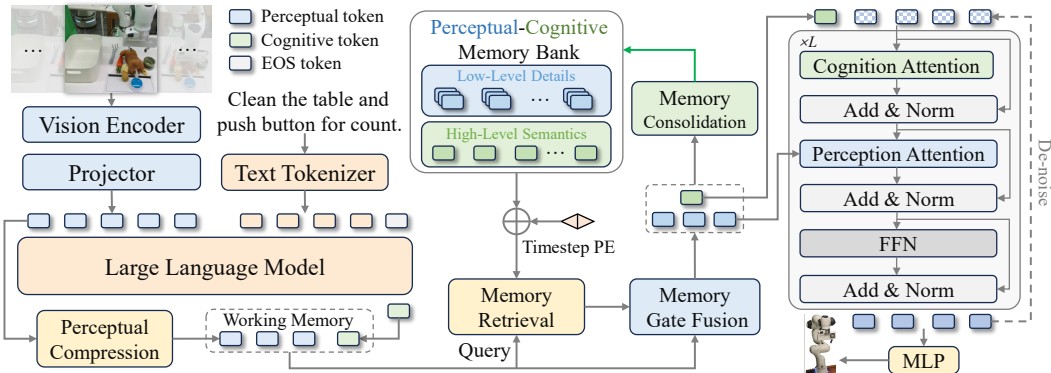

Figure 2: **Overall architecture of MemoryVLA.** RGB observation and language instruction are encoded by a 7B VLM into perceptual and cognitive tokens, forming short-term working memory. The working memory queries a perceptual-cognitive memory bank (PCMB) to retrieve relevant historical context, including high-level semantics and low-level visual details, adaptively fuses it with current tokens, and consolidates the PCMB by merging the most similar neighbors. The memory-augmented tokens then condition a diffusion transformer to predict a sequence of future actions.

## 3 METHOD

### 3.1 OVERVIEW OF MEMORYVLA

**Problem Formulation** We formulate robotic manipulation in VLA models as a sequential decision-making process, where visual observations and language instructions are mapped to control actions for real world interaction. Given the current RGB image $I \in \mathbb{R}^{H \times W \times 3}$ and a language instruction $L$, a parameterized policy $\pi$ outputs a sequence of future actions

$$\mathcal{A} = (a_1, \ldots, a_T) = \pi(I, L), \tag{1}$$

where each action $a_t = [\Delta x, \Delta y, \Delta z, \Delta \theta_x, \Delta \theta_y, \Delta \theta_z, g]^\top$ consists of relative translation, relative rotation (Euler angles), and a binary gripper state $g \in \{0, 1\}$.

**Overview** MemoryVLA is an end-to-end framework for robotic manipulation, as shown in Fig. 2. The current RGB observation and language instruction are first encoded by a VLM into perceptual and cognitive tokens, forming a working memory, analogous to neural activity in the visual and prefrontal cortex associated with short-term memory. To complement this short-term store, we introduce the Perceptual–Cognitive Memory Bank (PCMB), inspired by the hippocampus, which maintains long-term high-level semantics and fine-grained perceptual details. Working-memory embeddings query the PCMB to retrieve decision-relevant history, adaptively fuse it with current representations via gating, and consolidate the memory by merging temporally adjacent and semantically similar entries when capacity is reached. The resulting representations are then fed into a memory-conditioned diffusion action expert to generate a sequence of $N$ future 7-DoF actions.

### 3.2 VISION-LANGUAGE COGNITION MODULE

We build upon a 7B–parameter Prismatic VLM (Karamcheti et al., 2024), which is further pretrained on the large-scale cross-embodiment real robot dataset Open-X Embodiment (O'Neill et al., 2024). For visual encoding, we adopt parallel DINOv2 (Oquab et al., 2023) and SigLIP (Zhai et al., 2023) backbones on the current third-person RGB image $I$, concatenating their features into raw visual tokens. A perceptual compression module, implemented via a SE-bottleneck (Hu et al., 2018), then compresses these tokens into a compact set of perceptual tokens $p \in \mathbb{R}^{N_p \times d_p}$ with $N_p = 256$. In parallel, the raw visual tokens are projected via a linear layer into the language embedding space and concatenated with the tokenized instruction before being fed into the LLaMA-7B (Touvron et al., 2023). The output at the end-of-sentence (EOS) position is taken as the cognitive token $c \in \mathbb{R}^{1 \times d_c}$, representing high-level cognitive semantics in compact form. Finally, the perceptual tokens $p$ and cognitive token $c$ are combined to form the short-term working memory for downstream modules.

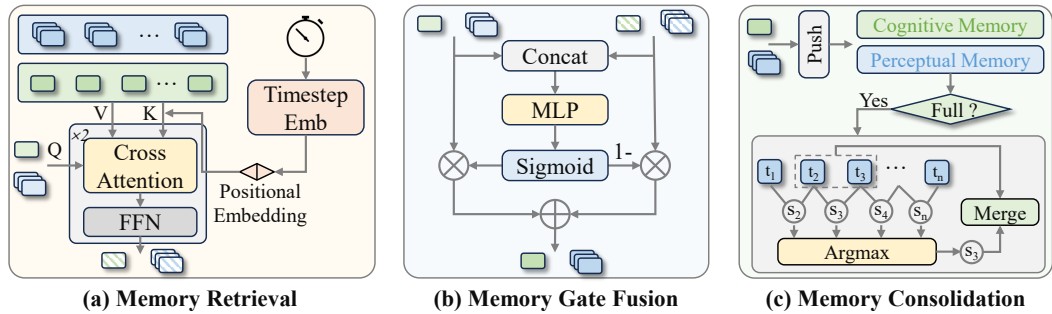

(a) Memory Retrieval     (b) Memory Gate Fusion     (c) Memory Consolidation

Figure 3: **Details of memory module.** (a) Retrieval: current perceptual and cognitive tokens query the PCMB via cross-attention with timestep positional encoding to fetch relevant historical features. (b) Gate fusion: current and retrieved tokens are adaptively fused via a gate mechanism. (c) Consolidation: the fused tokens are updated into PCMB. When PCMB reaches its capacity, we compute similarities between adjacent entries and merge the most similar pair to maintain compactness.

## 3.3 PERCEPTUAL-COGNITIVE MEMORY MODULE

The Vision–Language Cognition Module yields a working memory

$$M_{\mathrm{wk}} = \{\, p \in \mathbb{R}^{N_p \times d_p},\ c \in \mathbb{R}^{1 \times d_c} \,\}, \tag{2}$$

where $p$ and $c$ represent the current perceptual tokens and cognition token, respectively. However, this working memory only reflects the present timestep and lacks temporal dependencies.

To address this, inspired by the hippocampus in human memory systems, we introduce the Perceptual–Cognitive Memory Bank (PCMB):

$$M_{\mathrm{pcmb}} = \{\, m^x \mid x \in \{\mathrm{per}, \mathrm{cog}\} \,\}, \tag{3}$$

$$m^x = \{\, m^x_i \in \mathbb{R}^{N_x \times d_x} \,\}_{i=1}^{L}, \quad x \in \{\mathrm{per}, \mathrm{cog}\}, \tag{4}$$

where each perceptual entry $m^p_i$ stores fine-grained visual details and each cognitive entry $m^c_i$ encodes a high-level semantic summary. The bank maintains up to $L$ entries per stream.

**Memory Retrieval**     At each timestep, the working memory $M_{\mathrm{wk}}$, comprising current perceptual tokens $p \in \mathbb{R}^{N_p \times d_p}$ and cognition token $c \in \mathbb{R}^{1 \times d_c}$, acts as a dual query to retrieve historical information required for the current decision from the Perceptual-Cognitive Memory Bank $M_{\mathrm{pcmb}}$ as illustrated in Fig. 3 (a). Each memory entry is associated with its episode timestep via a sinusoidal embedding $\mathrm{TE}(\cdot)$, which is added as positional encoding. We then stack all perceptual memories into a tensor $\in \mathbb{R}^{L N_p \times d_p}$ and cognitive memories into a tensor $\in \mathbb{R}^{L \times d_c}$. Scaled dot-product attention between the current tokens and these memory tensors produces raw outputs for both streams:

$$K^x = [\, m^x_1 + \mathrm{TE}(t_1);\ \ldots;\ m^x_L + \mathrm{TE}(t_L)\,], \quad V^x = [\, m^x_1;\ \ldots;\ m^x_L\,], \tag{5}$$

$$\hat{H}^x = \mathrm{softmax}\!\left(\frac{q^x (K^x)^\top}{\sqrt{d_x}}\right) V^x, \quad q^x \in \{p, c\},\ x \in \{\mathrm{per}, \mathrm{cog}\}. \tag{6}$$

This attention operation is followed by a feed-forward network to complete one Transformer layer, and applying two such layers yields the final retrieved embeddings $H^p$ and $H^c$.

**Memory Gate Fusion**     As illustrated in Fig. 3 (b), the gate fusion process integrates the retrieved embeddings $H^p$ and $H^c$ with the current working memory representations through learned gates. For both the perceptual ($x = p$) and cognitive ($x = c$) streams, a gating vector is computed as

$$g^x = \sigma\big(\mathrm{MLP}(\mathrm{concat}[x,\ H^x])\big), \tag{7}$$

and applied to obtain the memory-augmented representation

$$\tilde{x} = g^x \odot H^x + (1 - g^x) \odot x. \tag{8}$$

Here, $\sigma$ denotes the sigmoid activation and $\odot$ denotes element-wise multiplication. The resulting memory-augmented features $\tilde{p}$ and $\tilde{c}$ are then forwarded to the memory consolidation stage.

**Memory Consolidation** After gate fusion, the memory-augmented representations $\tilde{p}$ and $\tilde{c}$ are passed to the Memory-conditioned Action Expert and simultaneously updated to the PCMB. When the number of stored entries exceeds $L$, cosine similarities are computed within each stream (perceptual and cognitive) between adjacent entries. The pair with the highest similarity in each stream is selected and merged by averaging their vectors, thereby reducing redundancy.

$$i_x^* = \arg\max_{i=1,\ldots,L-1} \cos(\tilde{x}_i, \tilde{x}_{i+1}), \quad m_{i_x^*}^x \leftarrow \tfrac{1}{2}(\tilde{x}_{i_x^*} + \tilde{x}_{i_x^*+1}), \quad x \in \{\text{per}, \text{cog}\}. \quad (9)$$

This consolidation mechanism (Fig. 3 (c)) mitigates memory bloat by reducing redundancy, while preserving the most salient perceptual details and semantic abstractions, thereby maintaining a compact representation that supports efficient long-term memory.

### 3.4 MEMORY-CONDITIONED ACTION EXPERT

Leveraging the memory-augmented working memory $\{\tilde{p}, \tilde{c}\}$, which integrates historical perceptual and cognitive information, the action expert predicts a sequence of future actions $\{a_1, a_2, \ldots, a_T\}$, with $T = 16$. This prediction allows the model to anticipate multi-step trajectories, reduce cumulative error, and provide foresight for long-horizon execution. Since real-world robotic actions lie in a continuous multimodal control space, we adopt a diffusion-based Transformer (DiT) (Peebles & Xie, 2023) implemented with Denoising Diffusion Implicit Models (DDIM) (Song et al., 2020), using 10 denoising steps for efficient yet accurate trajectory generation. This architecture progressively denoises a sequence of noisy action tokens, yielding precise continuous actions.

Specifically, at each denoising step, the noisy action tokens are injected with the sinusoidal encoding of the denoising timestep and concatenated with the cognitive representation $\tilde{c}$. A cognition-attention layer conditions the process with high-level semantic guidance, while a perception-attention layer supplements fine-grained visual details from the perceptual features $\tilde{p}$. The combined representation is then refined through a feed-forward network to obtain the denoised action at that step. The model is trained with mean squared error (MSE) loss between the predicted and target actions, and the final denoised vectors are passed through an MLP to generate continuous 7-DoF robotic actions.

## 4 EXPERIMENTS

To comprehensively evaluate MemoryVLA, we organize experiments around six core questions: (1) How does MemoryVLA compare with state-of-the-art methods on SimplerEnv benchmark? (Sec. 4.2) (2) How does it perform on LIBERO benchmark? (Sec. 4.3) (3) How does it perform on Mikasa-Robo benchmark? (Sec. 4.4) (4) Can it handle both general manipulation and long-horizon temporal tasks on real robots? (Sec. 4.5) (5) What is the impact of each component? (Sec. 4.6) (6) How robust and generalizable is it under diverse environmental conditions? (Appendix B)

### 4.1 EXPERIMENTAL SETUPS

**Simulation and Real-world Benchmarks.** Fig. 4 overviews our evaluation across simulation and real-world, covering 3 robots, 6 benchmarks, 150+ tasks with 500+ variations. SimplerEnv (Li et al., 2024b) includes Bridge suite with a WidowX robot and Fractal suite with a Google robot. Fractal provides two settings: *Visual Matching (VM)* and *Visual Aggregation (VA)*. LIBERO (Liu et al., 2023a) uses a Franka robot and spans five suites (Spatial, Object, Goal, Long-10, and Long-90). Mikasa-Robo (Cherepanov et al., 2025) uses a Franka robot. In real-world, we evaluate General and Long-horizon Temporal suites on Franka and WidowX robots. Task details for each benchmark and additional qualitative results are provided in Appendix K and Appendix M.

**Implementation Details** We train on 8 NVIDIA A100 GPUs with PyTorch FSDP, using 32 samples per GPU for a global batch of 256 and a learning rate of $2 \times 10^{-5}$. The model takes a single third-person RGB frame at $224 \times 224$ together with the language instruction and outputs 7-DoF actions. The LLM is 7B, and the diffusion action expert has $\sim$300M parameters. At inference we use DDIM (Song et al., 2020) with 10 sampling steps and a classifier-free guidance(CFG) (Ho & Salimans, 2022) guidance scale of 1.5. Additional details are provided in Appendix C and D.

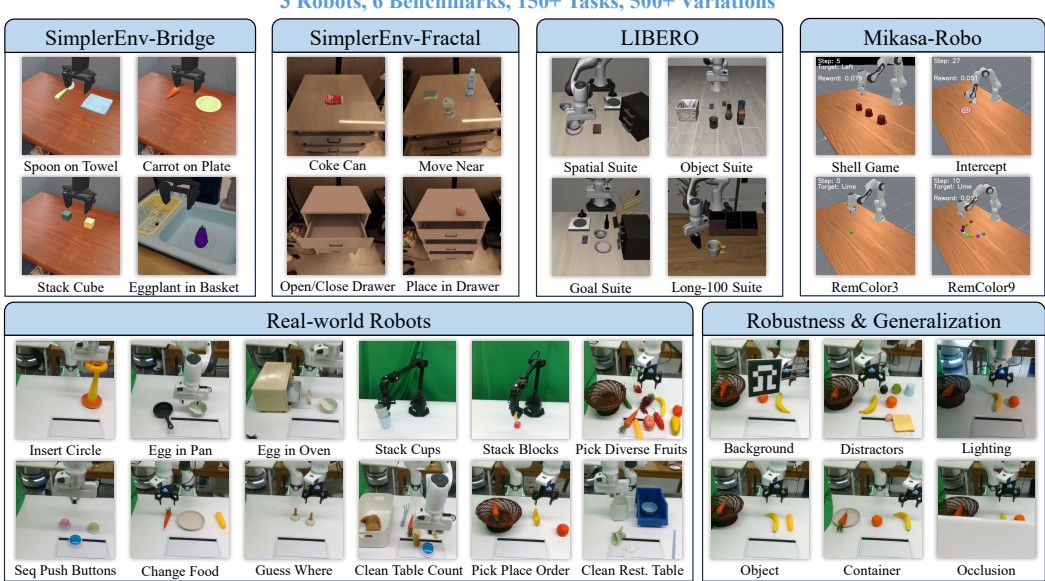

Figure 4: **Experimental setup overview.** Top: Four simulation benchmarks, SimpleEnv-Bridge, SimpleEnv-Fractal, LIBERO, and Mikasa-Robo. Bottom: real-world evaluation (General and Long-horizon Temporal), real-world robustness and generalization evaluation. In total, we evaluate 3 robots across 6 benchmarks, spanning over 150 tasks and 500 variations.

## 4.2 SIMULATED EVALUATION ON SIMPLERENV

**Training and Evaluation Setup** We evaluate on two SimplerEnv suites: Bridge and Fractal. For SimplerEnv-Bridge, we train on Bridge v2 dataset (Walke et al., 2023) for 50k steps with validation every 2.5k steps. Results are reported at the best validation step, and each task is evaluated with 24 trials to compute success rates. For SimplerEnv-Fractal, we train on the RT-1 dataset (Brohan et al., 2022) for 80k steps with validation every 5k steps. Evaluation covers *Visual Matching (VM)* and *Visual Aggregation (VA)* settings. VM mirrors the real setup to reduce sim-to-real gap, while VA stress-tests robustness by altering background, lighting, distractors, and table textures. The Fractal testbed includes 336 variants, yielding 2,856 trials in total.

**Evaluation Results on SimplerEnv-Bridge** As shown in Tab. 1, MemoryVLA achieves an average success rate of **71.9%**, a **+14.6** point gain over the CogACT-Large baseline, and surpasses recent state-of-the-art VLAs including $\pi_0$ (Black et al., 2024). Per task, success rates are 75.0% on *Spoon on Towel*, 75.0% on *Carrot on Plate*, 37.5% on *Stack Cube*, and 100.0% on *Eggplant in Basket*.

**Evaluation Results on SimplerEnv-Fractal** Tab. 2 reports results under *Visual Matching* and *Visual Aggregation* settings. MemoryVLA achieves an overall success rate of **72.7%**, improving CogACT by +4.6 points and surpassing $\pi_0$. By setting, the averages are **77.7%** on VM and **67.7%** on VA, gains of +2.9 and +6.4 points over CogACT, respectively. On *Open/Close Drawer* (VM), it reaches 84.7%, a +12.9 point improvement over CogACT; under VA we observe larger gains, including +24.9 on *Open/Close Drawer* and +11.7 on *Put in Drawer*.

## 4.3 SIMULATED EVALUATION ON LIBERO

**Training and Evaluation Setup** We evaluate on the LIBERO (Liu et al., 2023a) benchmark with a Franka robot across five suites: Spatial, Object, Goal, Long-10, and Long-90. The first four suites contain 10 tasks each, and Long-90 contains 90 tasks. Following OpenVLA (Kim et al., 2024), 50 demonstrations per task are used. Separate models are trained for Spatial, Object, and Goal for 20k steps each, while Long-10 and Long-90 are trained jointly for 40k steps. Validation is performed

Table 1: **Performance comparison on SimplerEnv-Bridge (Li et al., 2024b) with WidowX robot.** CogACT-Large is our re-evaluated baseline using official weight, and MemoryVLA achieves a +14.6 gain in average success. Entries marked with * are reproduced from open-pi-zero, which leverage additional proprioceptive state inputs; they also adopt Uniform/Beta timestep sampling.

| Method | Spoon on Towel | Carrot on Plate | Stack Cube | Eggplant in Basket | Avg. Success |
|---|---|---|---|---|---|
| RT-1-X (O'Neill et al., 2024) | 0.0 | 4.2 | 0.0 | 0.0 | 1.1 |
| OpenVLA (Kim et al., 2024) | 4.2 | 0.0 | 0.0 | 12.5 | 4.2 |
| Octo-Base (Team et al., 2024) | 15.8 | 12.5 | 0.0 | 41.7 | 17.5 |
| TraceVLA (Zheng et al., 2024b) | 12.5 | 16.6 | 16.6 | 65.0 | 27.7 |
| RoboVLMs (Liu et al., 2025b) | 45.8 | 20.8 | 4.2 | 79.2 | 37.5 |
| SpatialVLA (Qu et al., 2025) | 16.7 | 25.0 | 29.2 | 100.0 | 42.7 |
| Magma (Yang et al., 2025a) | 37.5 | 29.2 | 20.8 | 91.7 | 44.8 |
| CogACT-Base (Li et al., 2024a) | 71.7 | 50.8 | 15.0 | 67.5 | 51.3 |
| $\pi_0$-Uniform* (Black et al., 2024) | 63.3 | 58.8 | 21.3 | 79.2 | 55.7 |
| CogACT-Large (Li et al., 2024a) | 58.3 | 45.8 | 29.2 | 95.8 | 57.3 |
| CronusVLA (Li et al., 2025a) | 66.7 | 54.2 | 20.8 | 100.0 | 60.4 |
| $\pi_0$-Beta* (Black et al., 2024) | 84.6 | 55.8 | 47.9 | 85.4 | 68.4 |
| MemoryVLA (Ours) | 75.0 | 75.0 | 37.5 | 100.0 | **71.9** (+14.6) |

Table 2: **Performance comparison on SimplerEnv-Fractal (Li et al., 2024b) with Google robot.** Success rates (%) are reported for *Visual Matching (VM)* and *Visual Aggregation (VA)* suites. MemoryVLA achieves an overall +4.6 gain over CogACT. O./C. denotes Open/Close, and * follow Tab. 1.

| Method | Visual Matching (VM) | | | | | Visual Aggregation (VA) | | | | | Overall |
|---|---|---|---|---|---|---|---|---|---|---|---|
| | Coke Can | Move Near | O. / C. Drawer | Put in Drawer | Avg. | Coke Can | Move Near | O. / C. Drawer | Put in Drawer | Avg. | |
| Octo-Base (Team et al., 2024) | 17.0 | 4.2 | 22.7 | 0.0 | 11.0 | 0.6 | 3.1 | 1.1 | 0.0 | 1.2 | 6.1 |
| RT-1-X (O'Neill et al., 2024) | 56.7 | 31.7 | 59.7 | 21.3 | 42.4 | 49.0 | 32.3 | 29.4 | 10.1 | 30.2 | 36.3 |
| OpenVLA (Kim et al., 2024) | 18.0 | 56.3 | 63.0 | 0.0 | 34.3 | 60.8 | 67.7 | 28.8 | 0.0 | 39.3 | 36.8 |
| RoboVLMs (Liu et al., 2025b) | 76.3 | 79.0 | 44.9 | 27.8 | 57.0 | 50.7 | 62.5 | 10.3 | 0.0 | 30.9 | 44.0 |
| TraceVLA (Zheng et al., 2024b) | 45.0 | 63.8 | 63.1 | 11.1 | 45.8 | 64.3 | 60.6 | 61.6 | 12.5 | 49.8 | 47.8 |
| RT-2-X (O'Neill et al., 2024) | 78.7 | 77.9 | 25.0 | 3.7 | 46.3 | 82.3 | 79.2 | 35.5 | 20.6 | 54.4 | 50.4 |
| Magma (Yang et al., 2025a) | 75.0 | 53.0 | 58.9 | 8.3 | 48.8 | 68.6 | 78.5 | 59.0 | 24.0 | 57.5 | 53.2 |
| SpatialVLA (Qu et al., 2025) | 79.3 | 90.0 | 54.6 | 0.0 | 56.0 | 78.7 | 83.0 | 39.2 | 6.3 | 51.8 | 53.9 |
| $\pi_0$-Uniform* (Black et al., 2024) | 88.0 | 80.3 | 56.0 | 52.2 | 69.1 | – | – | – | – | – | – |
| $\pi_0$-Beta* (Black et al., 2024) | 97.9 | 78.7 | 62.3 | 46.6 | 71.4 | – | – | – | – | – | – |
| CogACT (Li et al., 2024a) | 91.3 | 85.0 | 71.8 | 50.9 | 74.8 | 89.6 | 80.8 | 28.3 | 46.6 | 61.3 | 68.1 |
| MemoryVLA (Ours) | 90.7 | 88.0 | 84.7 | 47.2 | **77.7** | 80.5 | 78.8 | 53.2 | 58.3 | **67.7** | **72.7** (+4.6) |

every 1k steps and results are reported at the best validation step. Each task is evaluated with 50 trials and per-suite average success rates are reported.

**Evaluation Results on LIBERO**  As shown in Tab. 3, MemoryVLA achieves an overall success rate of **96.5%**, improving CogACT by +3.3 points and surpassing $\pi_0$. Per-suite success rates are 98.4% on Spatial, 98.4% on Object, 96.4% on Goal, 93.4% on Long-10, and 95.6% on Long-90. Note that MemoryVLA uses only third-person RGB, without wrist views or proprioceptive states.

## 4.4 SIMULATED EVALUATION ON MIKASA-ROBO

**Training and Evaluation Setup**  We evaluate on the Mikasa-Robo (Cherepanov et al., 2025) benchmark. In our experiments, we compare MemoryVLA with VLA-style baselines used in Mikasa-Robo, including OpenVLA-OFT (Kim et al., 2025), PI-0 (Black et al., 2024), and SpatialVLA (Qu et al., 2025). We additionally reproduce CronusVLA (Li et al., 2025a), a contemporaneous VLA model that uses temporal context. Following the standard Mikasa-Robo protocol using 250 demonstrations per task at 128×128 resolution, 100 evaluation episodes per task and end-effector control. Standard 5 tasks are trained jointly for 20k steps, and validation is performed every 1k steps and results are reported at the best validation step.

Table 3: **Performance comparison on LIBERO (Liu et al., 2023a) with Franka robot.** Success rates (%) are reported across five suites. * indicates methods using additional proprioceptive and wrist-camera inputs. CogACT results are reproduced by us. For methods without LIBERO-90 results, we report the average over the first four suites.

| Method | Spatial | Object | Goal | Long-10 | Long-90 | Avg. Success |
|---|---|---|---|---|---|---|
| Diffusion Policy (Chi et al., 2023) | 78.3 | 92.5 | 68.3 | 50.5 | – | 72.4 |
| Octo (Team et al., 2024) | 78.9 | 85.7 | 84.6 | 51.1 | – | 75.1 |
| MDT (Reuss et al., 2024) | 78.5 | 87.5 | 73.5 | 64.8 | – | 76.1 |
| UniACT (Zheng et al., 2025b) | 77.0 | 87.0 | 77.0 | 70.0 | 73.0 | 76.8 |
| MaIL (Jia et al., 2024) | 74.3 | 90.1 | 81.8 | 78.6 | – | 83.5 |
| SpatialVLA (Qu et al., 2025) | 88.2 | 89.9 | 78.6 | 55.5 | 46.2 | 71.7 |
| TraceVLA (Zheng et al., 2024b) | 84.6 | 85.2 | 75.1 | 54.1 | – | 74.8 |
| OpenVLA (Kim et al., 2024) | 84.7 | 88.4 | 79.2 | 53.7 | 73.5 | 75.9 |
| CoT-VLA (Zhao et al., 2025) | 87.5 | 91.6 | 87.6 | 69.0 | – | 81.1 |
| $\pi_0$-FAST* (Pertsch et al., 2025) | 96.4 | 96.8 | 88.6 | 60.2 | 83.1 | 85.0 |
| CronusVLA (Li et al., 2025a) | 90.1 | 94.7 | 91.3 | 68.7 | | 86.2 |
| TriVLA (Liu et al., 2025d) | 91.2 | 93.8 | 89.8 | 73.2 | – | 87.0 |
| 4D-VLA (Zhang et al., 2025a) | 93.8 | 92.8 | 95.6 | 86.5 | – | 92.2 |
| CogACT (Li et al., 2024a) | 97.2 | 98.0 | 90.2 | 88.8 | 92.1 | 93.2 |
| $\pi_0$* (Black et al., 2024) | 96.8 | 98.8 | 95.8 | 85.2 | – | 94.2 |
| MemoryVLA (Ours) | 98.4 | 98.4 | 96.4 | 93.4 | 95.6 | **96.5** (+3.3) |

Table 4: **Performance comparison on Mikasa-Robo (Cherepanov et al., 2025) with Franka robot.** Success rates (%) are reported. CronusVLA results are reproduced by us.

| Model | ShellGame Touch | Intercept Medium | Remb. Color3 | Remb. Color5 | Remb. Color9 | Avg. Success |
|---|---|---|---|---|---|---|
| CronusVLA (Li et al., 2025a) | 32 | 5 | 31 | 13 | 9 | 18.0 |
| SpatialVLA (Qu et al., 2025) | 23 | 27 | 27 | 17 | 11 | 21.0 |
| OpenVLA-OFT (Kim et al., 2025) | 47 | 14 | 59 | 16 | 6 | 28.4 |
| PI-0 (Black et al., 2024) | 33 | 42 | 35 | 22 | 15 | 29.4 |
| MemoryVLA (Ours) | 88 | 24 | 44 | 30 | 20 | **41.2** (+11.8) |

**Evaluation Results on Mikasa-Robo** MemoryVLA consistently achieves the highest performance across all tasks, with an average of **11.8%** improvement over the previous state-of-the-art, especially **+41.0%** on ShellGameTouch task. The results are shown in Tab. 4.

## 4.5 REAL-WORLD EVALUATION

**Training and Evaluation Setup** We evaluate two real-robot suites, *General* and *Long-horizon Temporal*, on Franka and WidowX robots. Both use an Intel RealSense D435 RGB camera mounted in a fixed front view. Images are captured at $640 \times 480$ and downsampled to $224 \times 224$. The system is integrated via ROS. For *General*, each task uses 50-150 demonstrations and is evaluated from randomized initial states. *Pick Diverse Fruits* comprises five variants with 5 trials per variant (25 total); all other General tasks use 15 trials. For *Long-horizon Temporal*, each task uses 200-300 demonstrations and is evaluated with 10-15 trials using step-wise scoring to reflect progress over sub-goals. Training runs for approximately 5k–20k steps depending on the task and data size.

**Evaluation Results on Real-world** As shown in Tab. 5, MemoryVLA achieves average success scores of **85%** on general tasks and **83%** on long-horizon temporal tasks, exceeding CogACT by **+9** and **+26** percentage points, respectively, and surpassing $\pi_0$ across both suites. On general tasks, it exceeds the strongest baseline on every task, with notable gains on *Egg in Pan* (+13) and *Egg in Oven* (+20). On long-horizon temporal tasks, improvements are larger, including +43 on *Seq. Push Buttons*, +38 on *Change Food*, +32 on *Guess Where*, and +17 on *Clean Table & Count*. These results demonstrate strong real-world competence and highlight the benefits of temporal memory.

Table 5: **Performance comparison on real-world experiments with Franka and WidowX robots.** Success scores (%) are reported over six general tasks and six long-horizon temporal tasks. All methods are evaluated with only third-person RGB observation and language instruction.

| Method | General Tasks | | | | | | |
|---|---|---|---|---|---|---|---|
| | Insert Circle | Egg in Pan | Egg in Oven | Stack Cups | Stack Blocks | Pick Diverse Fruits | Avg. Success |
| OpenVLA (Kim et al., 2024) | 47 | 27 | 53 | 40 | 13 | 4 | 31 |
| $\pi_0$ (Black et al., 2024) | 67 | 73 | 73 | 87 | 53 | 80 | 72 |
| CogACT (Li et al., 2024a) | 80 | 67 | 60 | 93 | 80 | 76 | 76 |
| MemoryVLA (Ours) | 87 | 80 | 80 | 93 | 87 | 84 | **85** (+9) |

| Method | Long-horizon Temporal Tasks | | | | | | |
|---|---|---|---|---|---|---|---|
| | Seq. Push Buttons | Change Food | Guess Where | Clean Table & Count | Pick Place Order | Clean Rest. Table | Avg. Success |
| OpenVLA (Kim et al., 2024) | 6 | 3 | 0 | 15 | 27 | 0 | 9 |
| $\pi_0$ (Black et al., 2024) | 25 | 42 | 24 | 61 | 82 | 80 | 52 |
| CogACT (Li et al., 2024a) | 15 | 47 | 40 | 67 | 90 | 84 | 57 |
| MemoryVLA (Ours) | 58 | 85 | 72 | 84 | 100 | 96 | **83** (+26) |

Table 6: **Ablation on memory type and length.** We report average success rates (%) on SimplerEnv-Bridge tasks.

| | Variant | Avg. Success |
|---|---|---|
| Memory Type | Cognitive Mem. | 63.5 |
| | Perceptual Mem. | 64.6 |
| | Both | **71.9** |
| Memory Length | 4 | 67.7 |
| | 16 | **71.9** |
| | 64 | 67.7 |

Table 7: **Ablation on memory retrieval, fusion, consolidation.** We report average success rates (%) on SimplerEnv-Bridge tasks.

| | Variant | Avg. Success |
|---|---|---|
| Retrieval | w/o Timestep PE | 69.8 |
| | w/ Timestep PE | **71.9** |
| Fusion | Add | 67.7 |
| | Gate | **71.9** |
| Consolidation | FIFO | 66.7 |
| | Token Merge | **71.9** |

## 4.6 ABLATION STUDIES

We ablate memory design on SimplerEnv-Bridge to quantify each choice. As shown in Tab. 6, combining perceptual and cognitive memory attains 71.9%, compared with 63.5% for cognitive-only and 64.6% for perceptual-only. A memory length of 16 performs best at 71.9%, whereas 4 and 64 drop to 67.7%. Tab. 7 evaluates retrieval, fusion, and consolidation. Adding timestep positional encoding increases performance from 69.8% to 71.9%. Gate fusion reaches 71.9%, compared with 67.7% for simple addition. Token-merge consolidation achieves 71.9% versus 66.7% with FIFO.

## 5 CONCLUSION

Inspired by cognitive science, we propose MemoryVLA, a Cognition-Memory-Action framework for robotic manipulation. It uses a hippocampus-like Perceptual–Cognitive Memory Bank that cooperates with working memory to capture temporal dependencies. VLM commonsense priors further support high-level cognition, while a memory-conditioned diffusion action expert generates temporally aware actions. Across 150+ tasks with 500+ variations on 3 robots spanning SimplerEnv, LIBERO, and real-world, MemoryVLA consistently surpasses CogACT and $\pi_0$, achieves state-of-the-art performance, with notable gains on challenging long-horizon temporal tasks. It also demonstrates strong robustness and generalization under diverse OOD conditions. Future directions include (i) developing *memory reflection*, aligning long-term memory to the LLM input space to enable embedding-space chain-of-thought reasoning; and (ii) building *lifelong memory* through biologically inspired consolidation that distills frequently reused experiences into permanent representations, thereby supporting scalable generalization across scenes, tasks, and embodiments.

## ACKNOWLEDGMENTS

This work was supported by the National Science and Technology Major Project of China under Grant No. 2023ZD0121300, the National Natural Science Foundation of China under Grants U24B20173 and 62321005, the Scientific Research Innovation Capability Support Project for Young Faculty under Grant ZYGXQNJSKYCXNLZCXM-I20.

## REPRODUCIBILITY STATEMENT

The main paper specifies the model architectures, training setups, and experimental protocols, and the appendix provides additional details on hyperparameters, datasets, preprocessing and data augmentation, training procedures, evaluation methods and scoring rules, robustness and generalization analyses, task specifications, and qualitative results. For full reproducibility, we release:

- **Code**: https://github.com/shihao1895/MemoryVLA
- **Models**: https://huggingface.co/collections/shihao1895/memoryvla
- **Logs**: https://huggingface.co/collections/shihao1895/memoryvla
- **Robotic videos**: https://shihao1895.github.io/MemoryVLA

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

APPENDIX

## A  LLM USAGE

Large Language Models (LLMs) were used only for grammar and wording polishing. All research content and contributions are entirely the responsibility of the authors, with no involvement of LLMs.

## B  ROBUSTNESS AND GENERALIZATION EVALUATION

### B.1  REAL-WORLD EVALUATION

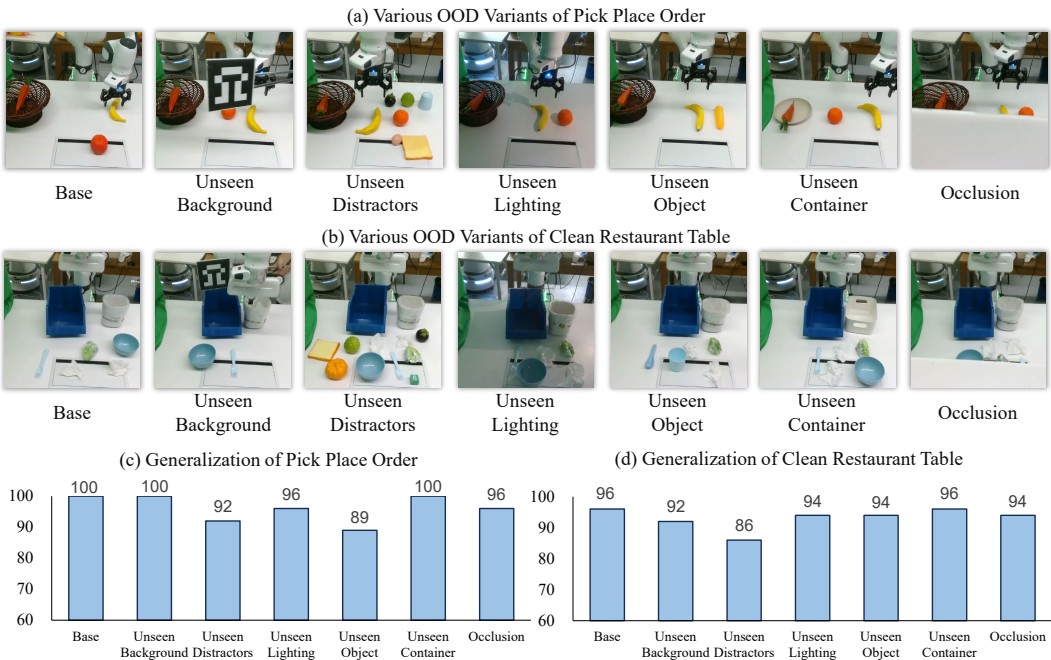

Figure 5: **Robustness and generalization under out-of-distribution (OOD) conditions in real-world.** (a,b) Examples of OOD variants for two representative tasks (*Pick Place Order* and *Clean Restaurant Table*), including unseen backgrounds, distractors, lighting, novel objects/containers, and occlusion. (c,d) Quantitative results showing that MemoryVLA maintains high success rates across these OOD variants, demonstrating strong robustness and generalization in real-world environments.

We further assess the robustness and generalization of MemoryVLA in real-world environments under diverse out-of-distribution (OOD) variants. Fig. 5 shows two representative tasks, *Pick Place Order* and *Clean Restaurant Table*, evaluated under unseen backgrounds, distractors, novel objects/containers, lighting variations, and occlusions.

For *Pick Place Order*, MemoryVLA attains near-perfect success under the base setting (100%), unseen background (100%), unseen distractors (92%), unseen lighting (96%), unseen container (100%), and occlusion (96%), with a moderate drop on unseen objects (89%). For *Clean Restaurant Table*, the base success rate is 96%, with unseen background (92%), unseen distractors (86%), unseen lighting (94%), unseen object (94%), unseen container (96%), and occlusion (94%).

These results confirm that MemoryVLA maintains consistently high performance across a wide range of real-world OOD conditions, demonstrating strong robustness and generalization.

### B.2  SIMULATION EVALUATION

We further conduct robustness and generalization experiments in simulation, considering both pick-and-move tasks and hinge-like object manipulation tasks. Fig. 6 presents results on *Pick Coke Can*

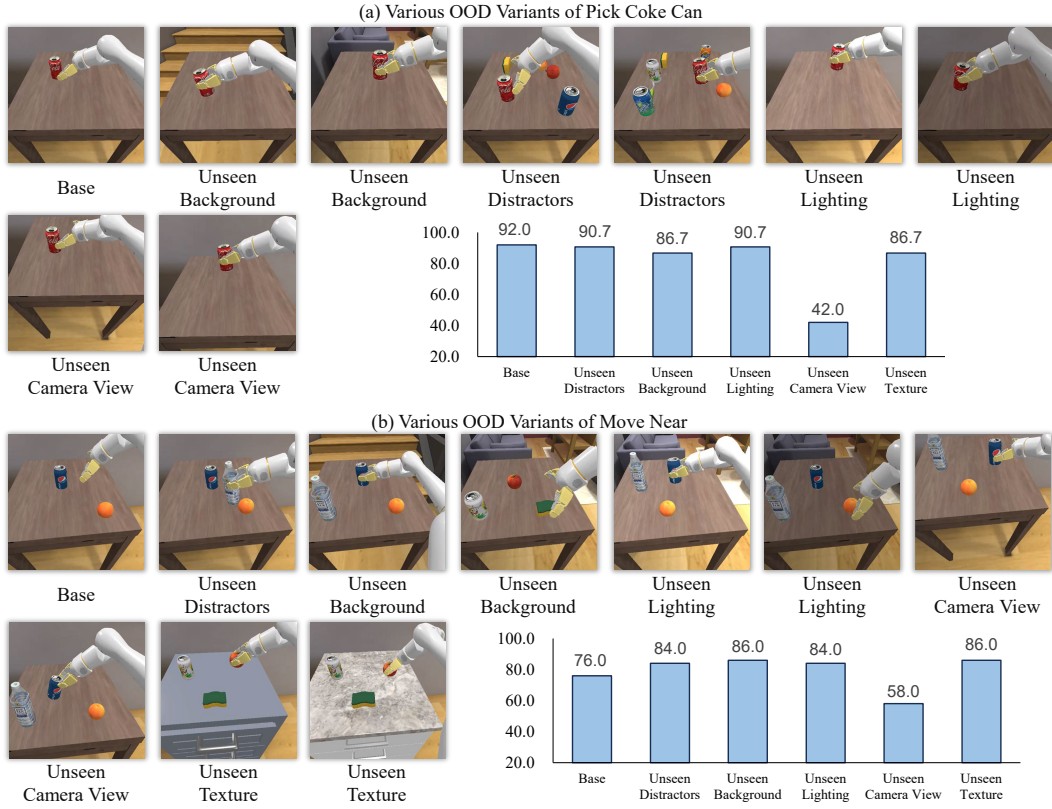

Figure 6: **Robustness and generalization under out-of-distribution (OOD) variants in simulation: Pick and Move tasks.** (a) *Pick Coke Can* and (b) *Move Near* tasks evaluated under unseen backgrounds, distractors, lighting, textures, and camera views. Bar plots report the corresponding success rates, showing that MemoryVLA maintains strong performance across most shifts, with the largest degradation under unseen camera views.

and *Move Near*, while Fig. 7 covers *Open/Close Drawer* and *Place Apple Into Drawer*. These tasks are evaluated under unseen backgrounds, distractors, lighting, textures, and camera views.

For *Pick Coke Can*, MemoryVLA achieves a base success rate of 92.0%, with unseen distractors (90.7%), unseen background (86.7%), unseen lighting (90.7%), and unseen texture (86.7%), while performance drops substantially under unseen camera views (42.0%). For *Move Near*, the base success rate is 76.0%, with unseen distractors (84.0%), unseen background (86.0%), unseen lighting (84.0%), unseen camera view (58.0%), and unseen texture (86.0%). For hinge-like object manipulation, *Open/Close Drawer* yields a base success rate of 46.3%, unseen background (56.4%), unseen lighting (49.1%), and unseen texture (57.4%). For *Place Apple Into Drawer*, the base success rate is 72.0%, with unseen background (66.0%), unseen lighting (52.0%), and unseen texture (50.0%).

These results show that MemoryVLA generalizes well across moderate distribution shifts such as distractors, backgrounds, and textures, but suffers more under severe changes, especially unseen camera views.

## C   ADDITIONAL TRAINING DETAILS

### C.1   HYPER-PARAMETERS

We summarize the main hyperparameters used in our experiments. The global batch size is 256 ($32 \times 8$ GPUs), the learning rate is $2 \times 10^{-5}$, and gradients are clipped at a max norm of 1.0. The policy predicts a 16-step action chunk, and perceptual tokens use 256 channels. The diffusion policy

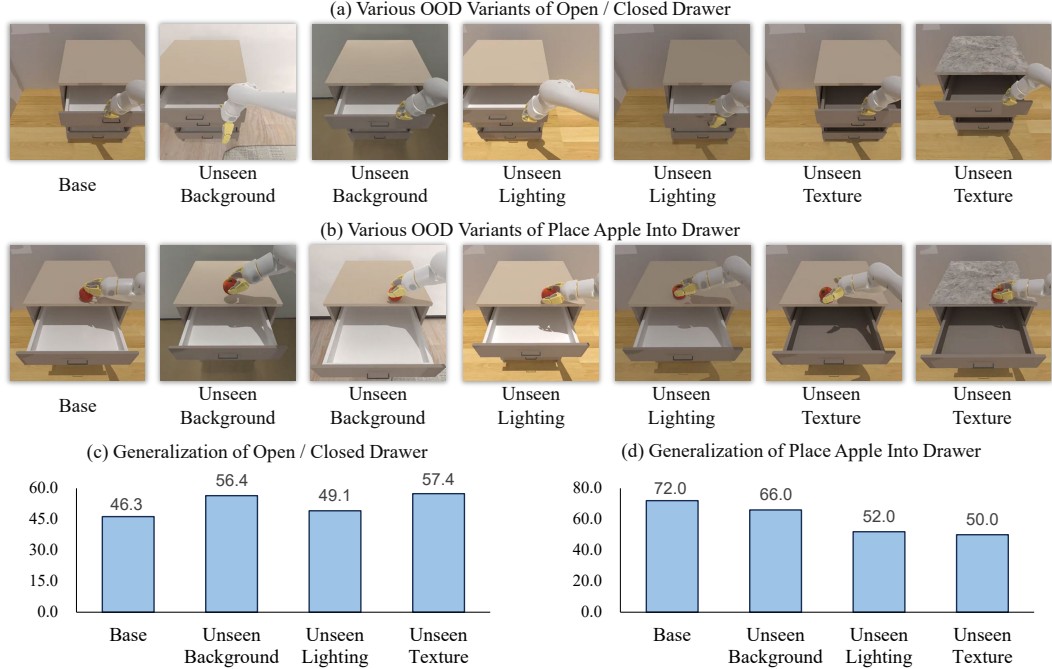

Figure 7: **Robustness and generalization under out-of-distribution (OOD) variants in simulation: Hinge-like object manipulation.** (a) OOD variants of *Open/Close Drawer* and (b) *Place Apple Into Drawer* tasks, including unseen backgrounds, distractors, lighting, textures, and camera views. Quantitative results indicate that MemoryVLA generalizes well under moderate shifts, while performance drops notably with camera views changes.

Table 8: **Training and model hyperparameters.**

| Hyperparameter | Value |
|---|---|
| Batch size | $32 \times 8$ |
| Learning rate | $2 \times 10^{-5}$ |
| Repeated diffusion steps | 4 |
| Action trunking size | 16 |
| Perceptual token channels | 256 |
| Max grad. norm | 1.0 |
| CFG scale (classifier-free guidance) | 1.5 |

uses 4 repeated diffusion steps during training; inference uses DDIM with 10 sampling steps and a classifier-free guidance scale of 1.5. See Tab. 8 for a concise summary.

## C.2 TRAINING DATA

**Bridge v2** For the SimplerEnv-Bridge benchmark, we train on BridgeData v2 (Walke et al., 2023), a large-scale, language-conditioned real-robot manipulation dataset of roughly 60,000 teleoperated trajectories collected on WidowX robots across diverse tabletop settings. Episodes pair language instructions with demonstrations of skills such as picking, placing, pushing, stacking, and folding.

**RT-1** For the SimplerEnv-Fractal benchmark, we use RT-1 (Brohan et al., 2022), a large-scale real-world dataset of roughly 130,000 episodes spanning 700+ tasks, collected over 17 months by the Google Robot fleet and paired with natural-language instructions.

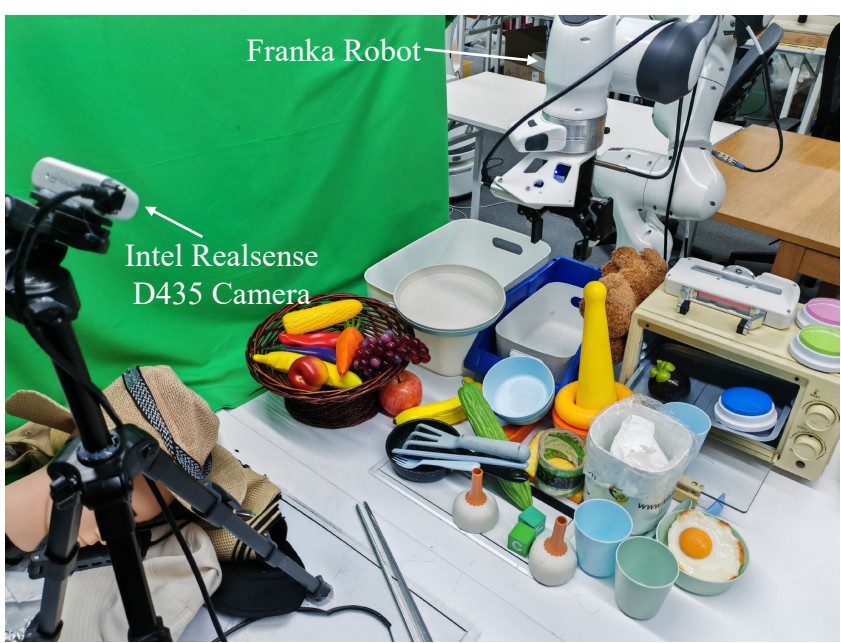

Figure 8: **Franka robot setup.**

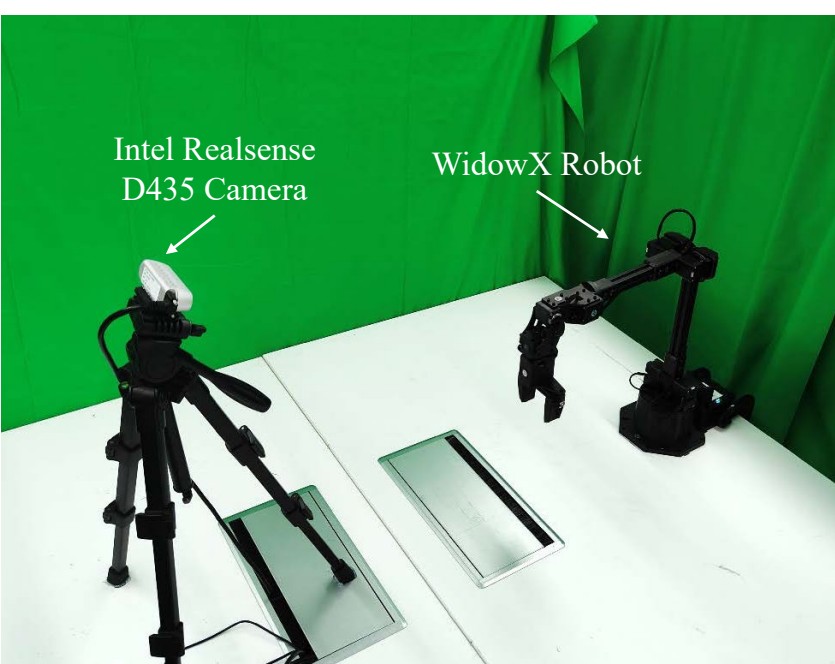

Figure 9: **WidowX robot setup.**

**LIBERO**  LIBERO (Liu et al., 2023a) provides simulation tasks with a Franka robot across five suites: Spatial, Object, Goal, Long-10, and Long-90, totaling 130 language-conditioned tasks. Each task supplies 50 demonstrations.

**Mikasa-Robo**  Mikasa-Robo (Cherepanov et al., 2025) comprises five memory-dependent manipulation tasks, each with 250 officially provided demonstrations, using $\Delta$ end-effector control.

**Real-world**   We collect real demonstrations on Franka and WidowX robots using a fixed third-person RGB setup, as shown in Fig. 8 and 9. A front-facing Intel RealSense D435 captures $640 \times 480$ RGB at 30 fps. Franka uses a single end-effector per experiment, either the stock parallel gripper or a Robotiq parallel gripper. Demonstrations are gathered by joystick teleoperation. The General suite uses 50-150 demonstrations per task, and the Long-horizon Temporal suite uses 200-300 per task. The system is integrated in ROS.

After collection, we perform a standardized preprocessing pipeline. Frames are downsampled to $224 \times 224$. We then subsample the video stream by retaining a frame whenever the end-effector translation since the last kept frame exceeds $0.01\,\mathrm{m}$ or the orientation change exceeds $0.4\,\mathrm{rad}$, and we also enforce a maximum gap of 120 frames between kept frames. The processed episodes are converted into the RLDS format for downstream training.

## C.3   Training Setup

**SimplerEnv-Bridge**   On Bridge v2, models are trained for 50k steps with a stream dataloader. Each episode is unpacked into consecutive frames tagged with its episode ID. During training, batches are filled sequentially with frames from a single episode whenever possible. If an episode ends before the batch is complete, the remaining slots are filled with frames from the following episode. A new batch then continues from the position where the previous one stopped, ensuring that in-episode temporal order is always preserved. The memory length is fixed to 16.

**SimplerEnv-Fractal**   Models are trained for 80k steps on RT-1. The benchmark defines two protocols: Visual Matching (VM), which mirrors the real-robot setup, and Visual Aggregation (VA), which perturbs background, lighting, distractors, and textures to test robustness. The dataloader design and memory length follow the same setup as in SimplerEnv-Bridge.

**LIBERO**   Following OpenVLA (Kim et al., 2024), we train with 50 demonstrations per task after removing failed trajectories from the dataset. Spatial, Object, and Goal suites are trained separately for 20k steps each, while Long-10 and Long-90 are treated as a single family of long-horizon data and trained jointly for 40k steps. The dataloader adopts a grouped sampling strategy: in each iteration, 16 frames are randomly sampled from within a single episode, matching the memory length of 16 used throughout training. The dataloader adopts a grouped sampling strategy in which each batch is divided into multiple groups, and each group consists of several frames drawn from a single episode. Frames within a group are kept in temporal order. The memory length is set to 16.

**Mikasa-Robo**   Following Mikasa-Robo (Cherepanov et al., 2025), we adopt the standard protocol with five tasks and train jointly on all 1,250 demonstrations for 20k steps, using $128 \times 128$ RGB observations and $\Delta$ end-effector control. We reuse the same dataloader setup as in LIBERO, and set the memory length to 16.

**Real-world**   Models are trained for 5k-20k steps depending on task and dataset size. The general tasks contain 50-150 demonstrations per task, while long-horizon temporal tasks use 200-300 demonstrations per task. The memory length is set to 16 for general tasks and 256 for long-horizon temporal tasks.

## C.4   Data Augmentation

We apply standard per-frame augmentations to the third-person RGB stream during training. Augmentations are applied in a fixed order: random resized crop, random brightness, random contrast, random saturation, and random hue. The crop samples $90\%$ of the image area with aspect ratio $1.0$ and resizes to $224 \times 224$. Brightness is perturbed with magnitude $0.2$, contrast and saturation are scaled in $[0.8, 1.2]$, and hue is shifted by up to $0.05$. All augmentations are disabled at evaluation.

## D ADDITIONAL EVALUATION DETAILS

### D.1 SIMPLERENV

Evaluation follows the official CogACT protocol (Li et al., 2024a). We adopt the same evaluation scripts and use the adaptive action ensemble strategy introduced in CogACT, with ensemble coefficient $\alpha = 0.1$, ensemble horizon set to 7 for Bridge and 2 for Fractal. For Bridge, models are trained for 50k steps and validated every 2.5k steps, since the denoising objective of diffusion models does not reliably indicate policy quality, we report success rates at the best validation step. For Fractal, training runs for 80k steps with validation every 5k steps, and evaluation covers 336 variants in total (Tab. 19), we similarly report success rates at the best validation step, VM and VA settings are evaluated separately.

Since the original paper only reported per-task success rates for CogACT-Base but not for CogACT-Large, we re-evaluated the released CogACT-Large checkpoint in our setup and report those numbers for fairness. For $\pi_0$, results are taken from the open-source reproduction open-pi-zero, which provides implementations with both *uniform* and *beta* timestep sampling strategies in flow matching. We report results under float32 precision as in the public release. Note that open-pi-zero does not provide numbers for the Fractal *Visual Aggregation* setting, and thus these are missing.

### D.2 LIBERO

Evaluation on LIBERO (Liu et al., 2023a) is conducted across all five suites (Spatial, Object, Goal, Long-10, and Long-90). Models are validated every 1k steps, and each task is evaluated with 50 trials. Success rates are reported at the best validation step. Unlike SimplerEnv, no action ensemble strategy is used in our LIBERO experiments.

CogACT results are reproduced using the official codebase for fairness. For $\pi_0$ and $\pi_0$-FAST, we adopt reported numbers, noting that both methods leverage additional wrist-camera views and proprioceptive states, while our approach relies solely on a single third-person RGB. Despite this difference in input modalities, our method consistently surpasses, achieving stronger performance without extra sensory inputs.

### D.3 MIKASA-ROBO

For Mikasa-Robo (Cherepanov et al., 2025), we evaluate with 100 episodes per task. Validation is performed every 1k training steps, and success rates are reported at the checkpoint with the best validation performance. As in LIBERO, we do not use any action ensemble strategy in our Mikasa-Robo experiments.

### D.4 REAL-WORLD

Evaluation uses 15-25 trials for General tasks and 10-15 trials for Long-horizon Temporal tasks. For General tasks, *Pick Diverse Fruits* contains five variants (apple, orange, banana, chili, grape), each evaluated with 5 trials (25 total). All other General tasks are evaluated with 15 trials each, and we report task-level success rates.

For Long-horizon Temporal tasks, *Seq. Push Buttons* includes three button orders (blue-pink-green, blue-green-pink, green-blue-pink), each tested with 5 trials. All other tasks are evaluated with 10 trials, and step-wise scoring is adopted to capture partial progress. The scoring rules are as follows:

- **Seq. Push Buttons:** pressing each correct button yields 30, with a bonus of 10 if all three are correct. Loose matching is allowed (slight contact counts as a press).
- **Change Food:** lifting and removing the initial food (30), grasping the new food (30), and placing it on the plate (30), with a 10 bonus for full success.
- **Guess Where:** grasping the cover (30), covering the block (30), and uncovering it (40).
- **Clean Table & Count:** five objects in total. For each object, clearing yields 10 points and pressing the counter yields 10. Small counting errors (incomplete press / one extra press) earn 5; major errors (missed count / multiple extras) earn 0. Empty grasps with clear counting intent incur a 5-point penalty.

- **Pick Place Order:** carrot, banana, and orange must be picked and placed in sequence. Each correct step earns 30, with a 10 bonus for full completion. Any order violation terminates the attempt.
- **Clean Restaurant Table:** five objects in total. Each correctly sorted into trash bin or storage bin scores 20. Misplacement earns 10, and merely lifting without correct placement earns 5.

# E    CASE STUDY OF MEMORY RETRIEVAL

To provide a direct view of how the memory mechanism functions, Fig. 10 visualizes the retrieved memory elements and their attention weights on the real-world and simulation tasks. The model consistently attends to past frames that resolve decision-relevant ambiguities absent from the current observation.

In the real-world Change Food task, after the first food item is placed aside, the current frame contains two food items on the table, making it impossible to determine from this single observation which one should be picked next. MemoryVLAtherefore attends strongly to the nearby frames reflecting the recent motion trend, as well as the last decisive frame before the ambiguity arises. In the Shell Game Touch task from Mikasa-Robo Simulation Benchmark, the robot is briefly shown the cube location before it is covered by cups. The model consistently attends to the initial revealing frames, which provide the only reliable cue for identifying the correct cup. These results demonstrate that MemoryVLAretrieves meaningful temporal cues essential for disambiguating the next action, rather than simply recalling redundant visual history.

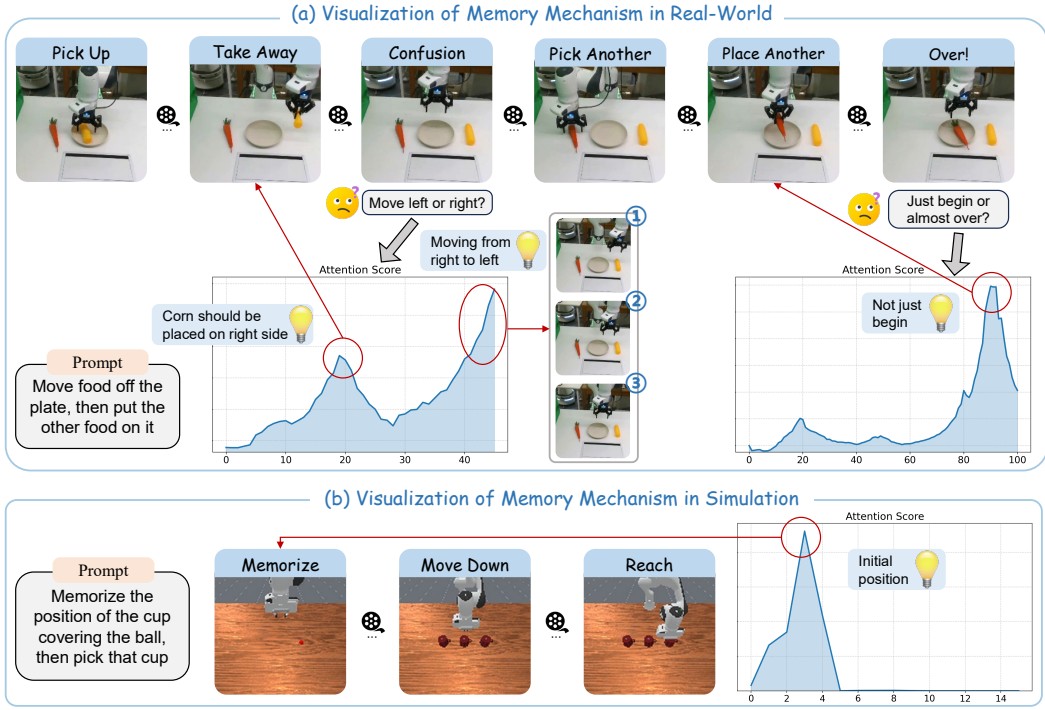

Figure 10: **Case study of memory retrieval in real-world and simulated tasks.** The figure visualizes the retrieved memory elements and their attention weights on the real-world *Change Food* task (top) and the simulated *Shell Game Touch* task in Mikasa-Robo (bottom). In both settings, the model consistently attends to past frames that resolve decision-relevant ambiguities absent from the current observation.

# F  Data Length Statistics

Tab. 9 reports the maximum, minimum, median, and average action lengths across all task suites, including SimplerEnv Evaluation (Bridge, Fractal), LIBERO (Spatial/Object/Goal and 10/90 task suites), and both real-world general and temporal tasks. For the real-world tasks, we additionally provide filtered statistics based on a motion-magnitude threshold (translation > 1 cm or rotation > 0.4 rad between consecutive frames) to remove frames where the end-effector motion is small.

Table 9: **Action Length Statistics** across all simulation (SimplerEnv Bridge/Fractal, LIBERO Spatial/Object/Goal, LIBERO-10/90) and real-world (General, Temporal) task suites. For real-world tasks, the "Filtered" versions remove frames whose end-effector motion is negligible (translation < 1 cm and rotation < 0.4 rad).

| Task Suite | Max | Min | Median | Average |
|---|---|---|---|---|
| SimplerEnv-Bridge / Fractal | 200 | 80 | 117 | 119 |
| LIBERO-Spatial / Object / Goal | 270 | 75 | 131 | 130 |
| LIBERO-10 / 90 | 505 | 58 | 144 | 156 |
| Real-General (Original) | 1575 | 281 | 575 | 575 |
| Real-General (Filtered) | 213 | 40 | 81 | 84 |
| Real-Temporal (Original) | 7704 | 412 | 981 | 1672 |
| Real-Temporal (Filtered) | 902 | 72 | 236 | 288 |

# G  Additional Ablation Study

For memory length, it is closely tied to the episode length of each benchmark. As shown in Tab. 9, SimplerEnv, LIBERO, and our real-world general tasks have similar episode lengths, whereas real-world temporal tasks are substantially longer, even after heavy frame filtering. This naturally motivates using a larger memory length for temporal tasks. We further conducted an ablation on a representative real-world temporal task (Clean Table & Count). A moderate memory length (256) performs best, as shown in Tab. 10.

We also added an ablation on the number of cognitive tokens. As shown in Tab. 11, increasing the count from 1 to 4 brings no performance gain. Perhaps the single 4096-dim EOS token already captures sufficient semantic information, so adding more tokens does not provide additional benefit.

In addition, we extended the ablations on fusion type (Tab. 12) and consolidation strategy (Tab. 13) to LIBERO-Long-90 and a real-world long-horizon task.

Table 10: **Additional ablation on memory length for both real-world and LIBERO-90 tasks.**

(a) Real-World: Clean Table & Count

| Memory Length | Success Rate |
|---|---|
| 64 | 78 |
| 256 (Base) | **84** |
| 512 | 81 |

(b) LIBERO-Long-90 Tasks

| Memory Length | Success Rate |
|---|---|
| 8 | 94.2 |
| 16 (Base) | **95.6** |
| 32 | 95.6 |

Table 11: **Ablation on the Number of Cognitive Tokens.** Increasing the number of cognitive tokens from 1 to 4 does not improve performance. A single 4096-dim EOS token already provides sufficient semantic capacity.

| Num. Cog Token | Spoon on Towel | Carrot on Plate | Stack Cube | Eggplant in Basket | Avg. Success |
|---|---|---|---|---|---|
| 1 (Base) | 75.0 | 75.0 | 37.5 | 100.0 | **71.9** |
| 4 | 79.2 | 66.7 | 37.5 | 95.8 | 69.8 |

Table 12: **Additional ablation on memory fusion type for both real-world and LIBERO-90.**

| Fusion Type | Clean Table & Count (Real-World) | LIBERO-Long-90 Tasks |
|---|---|---|
| Add | 78 | 93.8 |
| Gate | **84** | **95.6** |

Table 13: **Additional ablation on memory consolidation for both real-world and LIBERO-90.**

| Consolidation Type | Clean Table & Count (Real-World) | LIBERO-Long-90 Tasks |
|---|---|---|
| FIFO | 76 | 94.9 |
| Token Merge | **84** | **95.6** |

Tab. 14 provides an extended version of Tab. 6 and 7, reporting per-task success rates on SimplerEnv-Bridge for all ablation settings. Gray rows indicate the default configuration.

Table 14: **Details of ablation studies.** We report average success rates (%) on SimplerEnv-Bridge when varying five factors: (a) memory type, (b) memory length, (c) memory retrieval, (d) memory fusion, and (e) memory consolidation. Gray rows indicate the default configuration.

| Method | | Spoon on Towel | Carrot on Plate | Stack Cube | Eggplant in Basket | Avg. Success |
|---|---|---|---|---|---|---|
| (a) Memory Type | Cog. Mem. | 70.8 | 58.3 | 29.2 | 95.8 | 63.5 |
| | Per. Mem. | 83.3 | 54.2 | 20.8 | 100.0 | 64.6 |
| | Both | 75.0 | 75.0 | 37.5 | 100.0 | **71.9** |
| (b) Memory Length | 4 | 79.2 | 75.0 | 25.0 | 91.7 | 67.7 |
| | 16 | 75.0 | 75.0 | 37.5 | 100.0 | **71.9** |
| | 64 | 79.2 | 54.2 | 37.5 | 100.0 | 67.7 |
| (c) Memory Retrieval | w/o Timestep PE | 83.3 | 62.5 | 50.0 | 83.3 | 69.8 |
| | w/ Timestep PE | 75.0 | 75.0 | 37.5 | 100.0 | **71.9** |
| (d) Memory Fusion | Add | 75.0 | 62.5 | 33.3 | 100.0 | 67.7 |
| | Gate | 75.0 | 75.0 | 37.5 | 100.0 | **71.9** |
| (e) Memory Update | FIFO | 66.7 | 66.7 | 33.3 | 100.0 | 66.7 |
| | Token Merge | 75.0 | 75.0 | 37.5 | 100.0 | **71.9** |

## H    INFERENCE EFFICIENCY

As shown in Tab. 15, latency, throughput, and GPU memory measurements comparing our method with the baseline on two commonly used GPUs, RTX 4090 and HGX H20. The results below are averaged over 300 runs in bfloat16.

Our method achieves a latency of 0.194 s and a throughput of 82.5 Hz on RTX 4090, corresponding to only a 3.6% increase in overhead compared to the baseline. The memory usage is 16.6 GB, which is merely +0.8 GB over the baseline. These results show that the memory module is lightweight and introduces only negligible extra cost during inference.

The overhead remains small because the memory module is deliberately lightweight: each cognitive memory entry is represented by a single cognitive token, the perceptual memory is compressed to 256 channels through a perceptual compression module, and the memory consolidation step continuously merges redundant entries. Together, these designs keep both the retrieved memory size and cross-attention cost minimal, resulting in negligible additional latency and memory usage during inference.

Table 15: **Inference efficiency comparison.** Latency, throughput, and GPU memory are measured over 300 runs in bfloat16 with action chunk length set to 16 on RTX 4090 and HGX H20 GPU. MemoryVLA introduces only minor overhead compared to the baseline.

| Model | Latency (RTX 4090) | Throughput (RTX 4090) | Latency (HGX H20) | Throughput (HGX H20) | Memory |
|---|---|---|---|---|---|
| Baseline | 0.187 s | 85.6 Hz | 0.236 s | 67.8 Hz | 15.8 GB |
| MemoryVLA | 0.194 s | 82.5 Hz | 0.246 s | 65.0 Hz | 16.6 GB |

## I ZERO-SHOT TASK GENERALIZATION

In addition to visual OOD tests, we have added task generalization experiments to evaluate zero shot performance on unseen task categories. As shown in Tab. 16, We use Apple To Basket as the base task and test on three unseen tasks: Eggplant To Basket, Blush To Basket, and Apple To Plate. As shown in the table below, our method achieves good zero shot task generalization.

Table 16: **Zero-shot task generalization results.** Apple To Basket is used as the base task, and evaluation is conducted on three unseen task categories.

| Model | Apple To Basket (Base Task) | Eggplant To Basket (OOD Task) | Blush To Basket (OOD Task) | Apple To Plate (OOD Task) | Avg. Success (OOD Task) |
|---|---|---|---|---|---|
| MemoryVLA | 4 / 5 | 3 / 5 | 2 / 5 | 2 / 5 | **0.47** |

## J COMPARE WITH TEMPORAL CONTEXT VLA BASELINES

As shown in Tab. 17, we have added temporal-context baselines for each benchmark, including Mikasa-Robo (Cherepanov et al., 2025), LIBERO (Liu et al., 2023a), and SimplerEnv (Li et al., 2024b), to provide a fair and comprehensive comparison.

In the Mikasa-Robo benchmark, a memory-dependent manipulation benchmark, no temporal-context VLA baselines are provided in the official release. To ensure a fair comparison, we therefore reproduced CronusVLA (Li et al., 2025a), a contemporaneous strong VLA model that explicitly leverages historical information by aggregating multi-frame VLM features through a sliding-window module. This provides Mikasa-Robo with a temporal-context baseline, and MemoryVLA-substantially outperforms it across all tasks.

For the LIBERO benchmark, we include the main temporal-context VLA baselines commonly used in prior work, including TTF-VLA (Liu et al., 2025a), TraceVLA (Zheng et al., 2024b), 4D-VLA (Zhang et al., 2025a), CronusVLA (Li et al., 2025a), and MAP-VLA (Li et al., 2025b). These methods represent the strongest publicly available temporal-context VLA models that have reported results on LIBERO. MemoryVLAobtains higher average success rates than these baselines.

For the SimplerEnv benchmark, we include the temporal-context VLA baselines reported in prior work, including TraceVLA (Zheng et al., 2024b), RoboVLMs (Liu et al., 2025b), and CronusVLA (Li et al., 2025a). These methods incorporate explicit temporal modeling, and MemoryVLAachieves higher success rates than these baselines.

## K TASK DETAILS

To ensure comprehensive evaluation across simulation and real-world settings, we summarize the task design of each benchmark. We provide task templates, variation types, and the number of variations per task to clarify the diversity and difficulty of evaluation.

Table 17: **Comparison with temporal-context VLA methods across diverse benchmarks.** MemoryVLA outperforms temporal-context VLA baselines across both memory-focused benchmarks and general manipulation benchmarks, including Mikasa-Robo, SimplerEnv, and LIBERO.

(a) SimplerEnv Benchmark

| Method | Spoon on Towel | Carrot on Plate | Stack Cube | Eggplant in Basket | Avg. Success |
|---|---|---|---|---|---|
| TraceVLA (Zheng et al., 2024b) | 12.5 | 16.6 | 16.6 | 65.0 | 27.7 |
| RoboVLMs (Liu et al., 2025b) | 45.8 | 20.8 | 4.2 | 79.2 | 37.5 |
| CronusVLA (Li et al., 2025a) | 66.7 | 54.2 | 20.8 | 100.0 | 60.4 |
| MemoryVLA (Ours) | 75.0 | 75.0 | 37.5 | 100.0 | **71.9** |

(b) LIBERO Benchmark

| Method | Spatial | Object | Goal | Long-10 | Avg. Success |
|---|---|---|---|---|---|
| TTF-VLA (Liu et al., 2025a) | 73.0 | 84.0 | 81.0 | 58.0 | 74.0 |
| TraceVLA (Zheng et al., 2024b) | 84.6 | 85.2 | 75.1 | 54.1 | 74.8 |
| 4D-VLA (Zhang et al., 2025a) | 88.9 | 95.2 | 90.9 | 79.1 | 88.6 |
| CronusVLA (Li et al., 2025a) | 93.8 | 92.8 | 95.6 | 86.5 | 92.2 |
| MAP-VLA (Li et al., 2025b) | 96.3 | 98.4 | 95.4 | 83.4 | 93.4 |
| MemoryVLA (Ours) | 98.4 | 98.4 | 96.4 | 93.4 | **96.7** |

(c) Mikasa-Robo Benchmark

| Model | Shell Game Touch | Intercept Medium | Remb. Color3 | Remb. Color5 | Remb. Color9 | Avg. Success |
|---|---|---|---|---|---|---|
| CronusVLA (Li et al., 2025a) | 32 | 5 | 31 | 13 | 9 | 18.0 |
| MemoryVLA (Ours) | 88 | 24 | 44 | 30 | 20 | **41.2** |

Table 18: **Real-world tasks details.** We list the instruction template, number of variations, and the corresponding variation types for each task.

| Task Name | Language Instruction Template | # Variations | Variation Type |
|---|---|---|---|
| Seq Push Buttons | "Push the {color 1, color 2, and color 3} buttons in sequence" | 3 | Button color order |
| Change Food | "Move food off the plate, then put the other food on it" | 2 | Food object type in plate |
| Guess Where | "Place a cover over the block, then remove the cover" | 1 | – |
| Clean Table & Count | "Clean the table item by item, and push the button after each item cleaned" | 1 | – |
| Pick Place Order | "Pick up carrot, banana and orange in order and place them in the basket" | 7 | Background, distractors, lighting, object, container, occlusion |
| Clean Restaurant Table | "Place all trash into the trash bin and all tableware into the storage bin" | 7 | Background, distractors, lighting, object, container, occlusion |
| Insert Circle | "Insert the circle on the square" | 1 | – |
| Egg In Pan | "Put the egg into the pan" | 1 | – |
| Egg In Oven | "Put the egg into the oven" | 1 | – |
| Stack Cup | "Stack the green cup on the other cup" | 1 | – |
| Stack Block | "Stack the yellow block on the red block" | 1 | – |
| Pick Diverse Fruit | "Pick up {fruit} and place it in the basket" | 5 | Fruit category |

Table 19: **SimplerEnv tasks details.** *VM* = visual-matching, *VA* = variant-aggregation. For *Fractal*, we report *VM* and *VA* separately; all *Bridge* tasks use a single setting.

| | Task Name | Language Instruction Template | # Variations | Variation Type |
|---|---|---|---|---|
| *Bridge* | Spoon On Tower | "Put the spoon on the tower" | 1 | – |
| | Carrot On Plate | "Put the carrot on the plate" | 1 | – |
| | Stack Cube | "Stack the green cube on the yellow cube" | 1 | – |
| | Eggplant In Basket | "Put the eggplant in the basket" | 1 | – |
| *Fractal* | Pick Coke Can | "Pick up the coke can" | VM: 12 | VM: can position×3, URDF×4. |
| | | | VA: 33 | VA: base, backgrounds×2, lighting×2, textures×2, camera views×2, distractors×2, can position×3. |
| | Move Near | "Move the {object} near the {reference}" | VM: 4 | VM: URDF×4. |
| | | | VA: 10 | VA: base, distractors, backgrounds×2, lighting×2, textures×2, camera views×2. |
| | Open/Close Drawer | "Open/close the {level} drawer" | VM: 216 | VM: env {open/closed}×{top, middle, bottom}; URDF×4; initial pose×9. |
| | | | VA: 42 | VA: base, backgrounds×2, lighting×2, cabinet styles×2, env×6. |
| | Put In Drawer | "Put the {object} into the {level} drawer" | VM: 12 | VM: URDF×4; initial pose×3. |
| | | | VA: 7 | VA: base, backgrounds×2, lighting×2, cabinet styles×2. |

## K.1 REAL-WORLD TASKS

Tab. 18 shows the 12 tasks used in real-world evaluation, divided into *General* and *Long-horizon Temporal* suites.

**General Tasks.** *Insert Circle*: insert a circle onto a vertical pillar, requiring accurate positioning and insertion. *Egg in Pan*: place an egg into a shallow frying pan, testing grasp stability and gentle placement. *Egg in Oven*: put an egg into a small oven container, involving more constrained placement than the pan. *Stack Cups*: stack one plastic cup on top of another, evaluating vertical alignment and balance. *Stack Blocks*: stack a yellow block on top of a red block, focusing on precise spatial alignment. *Pick Diverse Fruits*: pick a specified fruit from a tabletop with more than ten different fruit types and place it into a basket, testing semantic understanding, visual diversity, and instruction following.

**Long-horizon Temporal Tasks.** *Seq. Push Buttons*: push three buttons in a specified color sequence, stressing ordered memory and resistance to temporal confusion. *Change Food*: remove a food item from a plate and replace it with another, requiring multi-step sequencing and correct temporal ordering. *Guess Where*: cover a block with a container and later uncover it, testing reversible actions and consistent tracking over time. *Clean Table & Count*: clear items from the table one by one while pressing a counter button after each removal, combining manipulation with explicit progress monitoring. *Pick Place Order*: pick up carrot, banana, and orange in a fixed order and place them into a basket, enforcing sequence-sensitive planning under temporal dependencies. *Clean Restaurant Table*: sort table items by category, placing trash into a trash bin and tableware into a storage bin, representing a long-horizon task with semantic reasoning and complex multi-stage sequencing.

## K.2 SIMPLERENV TASKS

Tab. 19 summarizes the tasks in the SimplerEnv benchmark, which consists of two suites: Bridge and Fractal.

Table 20: **LIBERO tasks details.** We list the language instruction templates and the total number of tasks per suite.

| Suite | Language Instruction Templates | #Tasks |
|---|---|---|
| Spatial | pick up the *OBJ SPATIAL_REL* and place it on the *TARGET* | 10 |
| Object | pick up the *FOOD* and place it in the *CONTAINER* | 10 |
| Goal | open/close the *CONTAINER*
open the *DRAWER* and put the *OBJ* inside
put the *OBJ* on/in the *TARGET*
push the *OBJ* to the *POSITION* of the *TARGET*
turn on the *APPLIANCE* | 10 |
| Long-10 | put both *OBJ1* and *OBJ2* in the *CONTAINER*
turn on the *APPLIANCE* and put the *OBJ* on it
put the *OBJ* in the *CONTAINER/APPLIANCE* and close it
place *OBJ1* on *TARGET1* and *OBJ2* on *TARGET2*/at *REL* of *TARGET2*
pick up the *OBJ* and place it in the caddy *COMPARTMENT* | 10 |
| Long-90 | open/close *CONTAINER/APPLIANCE* [and put *OBJ* on/in it; optionally sequence with another open/close]
open *CONTAINER* and put *OBJ* in it
put/place *OBJ* in/on/under *TARGET* or at *REL_POS*
stack *OBJ1* on *OBJ2* [optionally place them in *CONTAINER*]
pick up *OBJ* and put it in *CONTAINER* (basket/tray)
place *MUG* on left/right *PLATE* or *BOOK* in caddy/on/under shelf
turn on/off *APPLIANCE* [optionally put *OBJ* on it] | 90 |

The *Bridge* suite contains four tabletop manipulation tasks on WidowX robot: *Spoon on Towel*, *Carrot on Plate*, *Stack Cube*, and *Eggplant in Basket*. Each task is paired with a single language template, focusing on object placement and stacking primitives.

The *Fractal* suite builds on RT-1 data with Google robot and defines four tasks: *Pick Coke Can*, *Move Near*, *Open/Close Drawer*, and *Put in Drawer*. Each task is evaluated under two protocols. *Visual Matching (VM)* mirrors the real-world setup by varying object positions and URDFs, ensuring alignment between simulation and deployment. *Visual Aggregation (VA)* introduces substantial visual perturbations, including changes in backgrounds, textures, lighting, distractors, and camera views, to stress-test robustness and generalization. Together, VM and VA yield 336 variants, producing 2,352 evaluation trials.

### K.3 LIBERO TASKS

Tab. 20 outlines the five suites of the LIBERO benchmark: Spatial, Object, Goal, Long-10, and Long-90. LIBERO-Spatial consists of tasks where the same object must be placed across varying target positions. LIBERO-Object focuses on handling diverse objects within a fixed scene layout. LIBERO-Goal contains heterogeneous operations such as opening containers, placing objects, or turning on appliances, performed in an unchanged environment. Long-10 introduces ten extended tasks that require multiple sub-goals across different scenes, while Long-90 expands this setting to ninety tasks, providing a substantially more challenging benchmark. In total, LIBERO offers 130 tasks in simulation with a Franka robot.

## L VISUALIZATION OF MEMORY DEPENDENCE IN REAL-WORLD TASKS

To clarify why real-world temporal tasks require memory, Fig. 11 highlights the key moments in several tasks where the correct action depends on past information rather than the current observation. Fig. 12 further provides a step-by-step example from the *Change Food* task, showing a case where the next action becomes ambiguous from a single frame and can only be resolved by recalling

earlier steps. These examples illustrate that the stronger gains on real-robot tasks stem from their inherently memory-dependent nature.

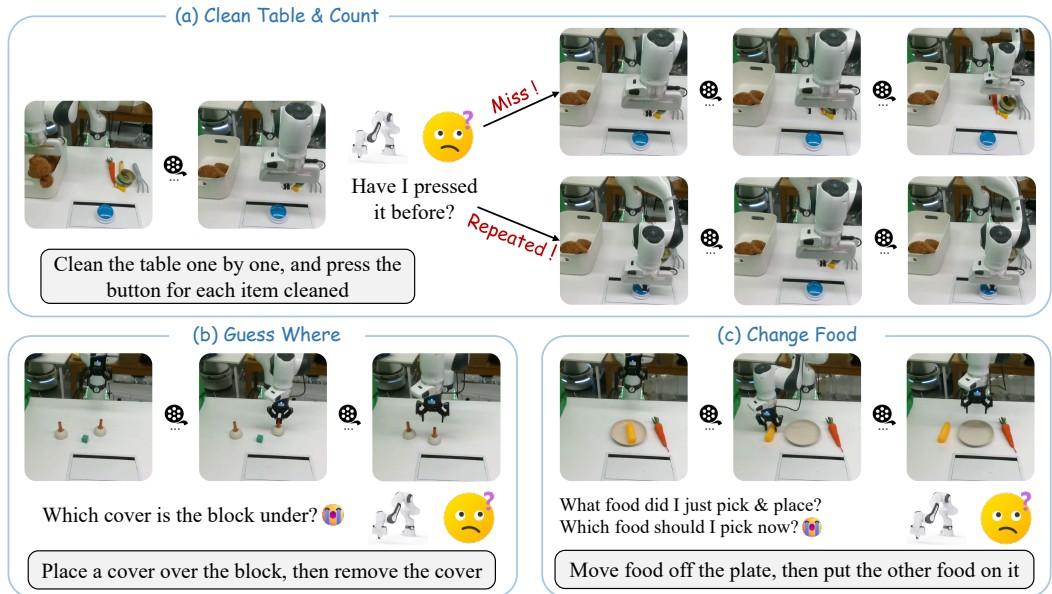

Figure 11: **Memory-dependent aspects of three real-world robotic tasks.** The tasks Clean Table & Count, Guess Where, and Change Food all require tracking past events to make correct decisions, highlighting their inherent temporal dependence.

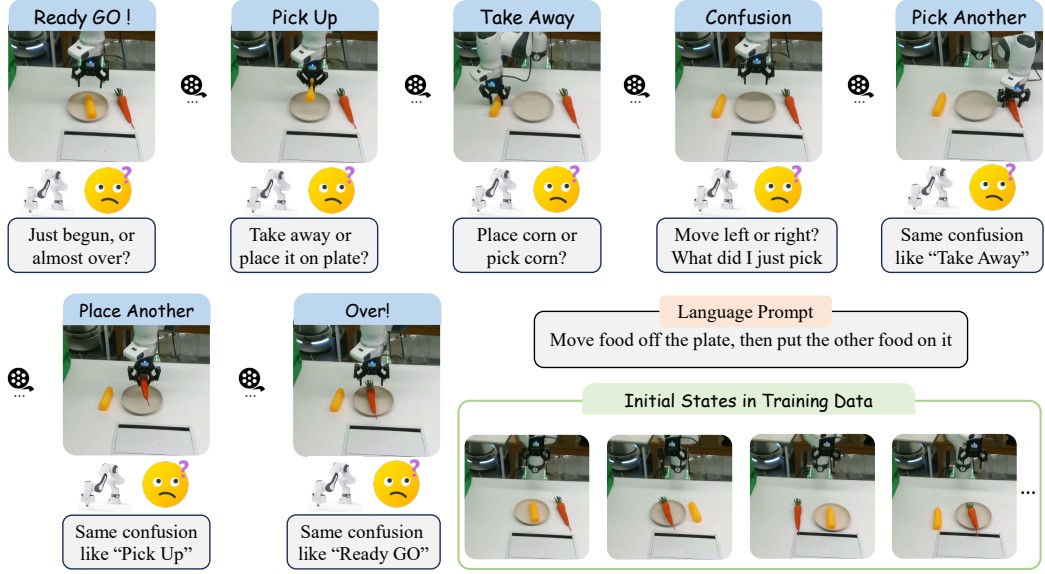

Figure 12: **Step-by-step example of memory-dependent behavior.** A sequence from the real-world Change Food task showing a case where the next action becomes ambiguous without recalling earlier steps.

# M QUALITATIVE RESULTS

## M.1 REAL-WORLD EVALUATION

We present qualitative examples to complement the quantitative evaluation. Fig. 13 and 14 illustrate rollouts on long-horizon temporal tasks in real-world. Fig. 15 shows general manipulation tasks in real-world.

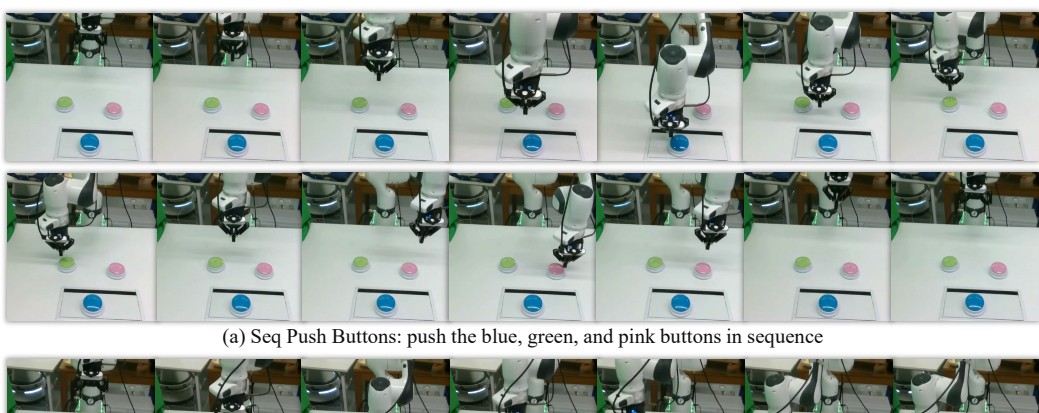

(a) Seq Push Buttons: push the blue, green, and pink buttons in sequence

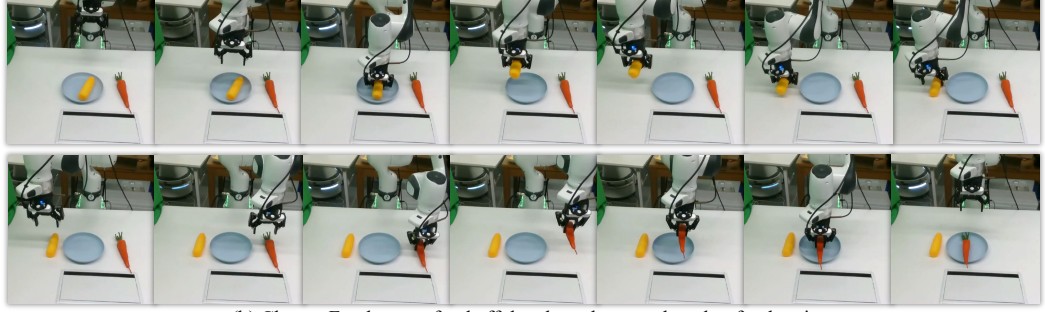

(b) Change Food: move food off the plate, then put the other food on it

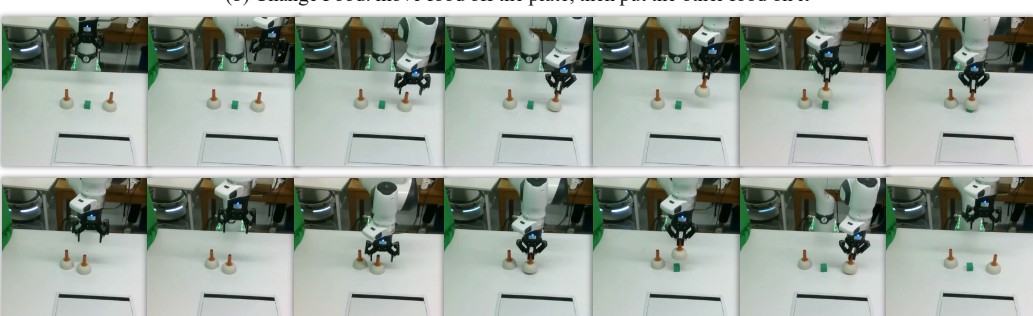

(c) Guess Where: place a cover over the block, then remove the cover

Figure 13: **Qualitative results of MemoryVLA on real-world long-horizon temporal tasks (I).** Representative examples include *Seq Push Buttons*, *Change Food*, and *Guess Where* tasks.

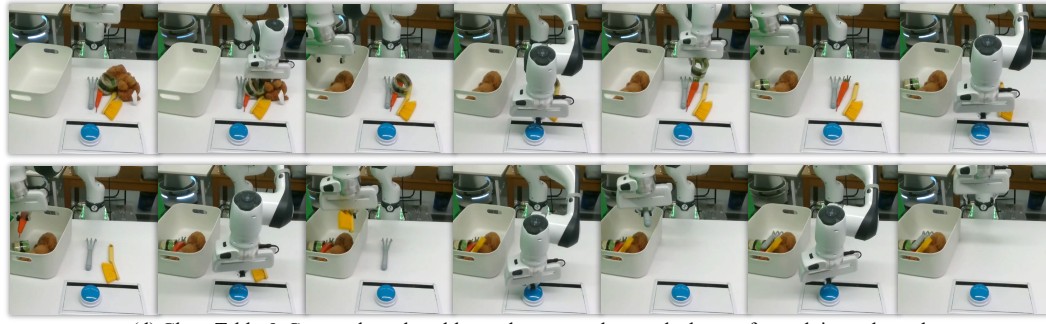

(d) Clean Table & Count: clean the table one by one, and press the button for each item cleaned

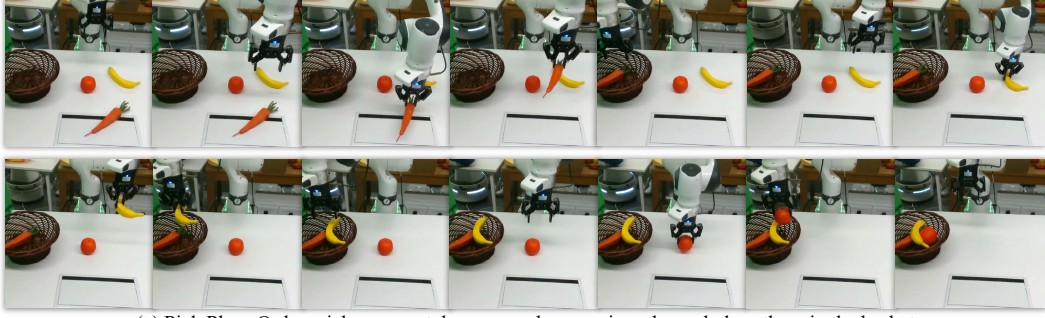

(e) Pick Place Order: pick up carrot, banana, and orange in order and place them in the basket

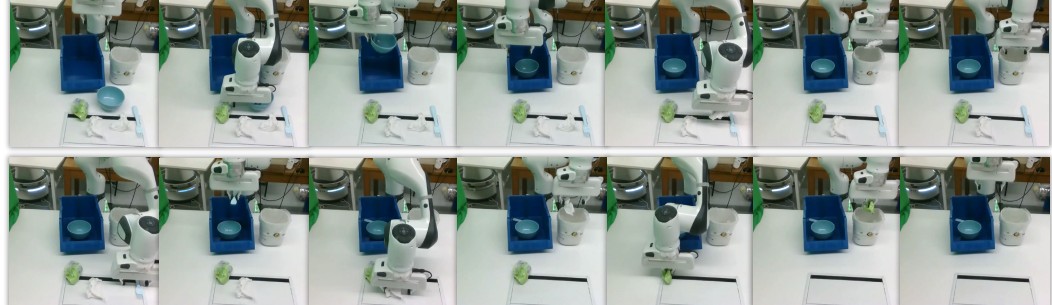

(f) Clean Restaurant Table: place all trash items into the trash bin and all tableware into the storage bin

Figure 14: **Qualitative results of MemoryVLA on real-world long-horizon temporal tasks (II).** Representative examples include *Clean Table & Count*, *Pick Place Order*, and *Clean Restaurant Table* tasks.

## M.2 SIMULATION EVALUATION

Results on simulated environments are visualized in Fig. 16 and 17, covering both Bridge and Fractal suites. Finally, Fig. 18 provides representative trajectories on LIBERO, spanning all five suites.

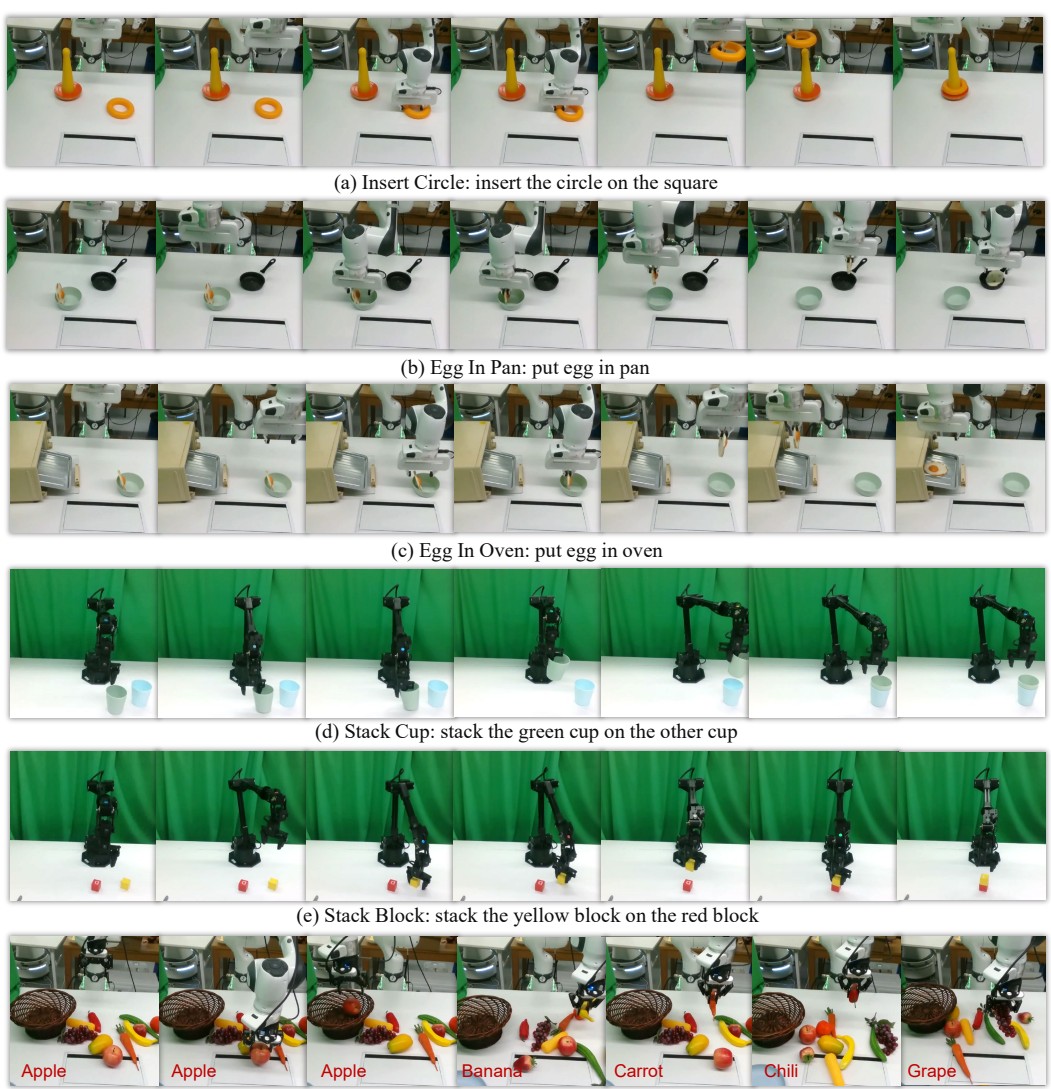

(a) Insert Circle: insert the circle on the square

(b) Egg In Pan: put egg in pan

(c) Egg In Oven: put egg in oven

(d) Stack Cup: stack the green cup on the other cup

(e) Stack Block: stack the yellow block on the red block

(f) Pick Diverse Fruit: pick up the apple and place it in the basket

Figure 15: **Qualitative results of MemoryVLA on real-world general tasks.** Representative examples include *Insert Circle*, *Egg in Pan*, *Egg in Oven*, *Stack Cups*, *Stack Blocks*, and *Pick Diverse Fruits* tasks.

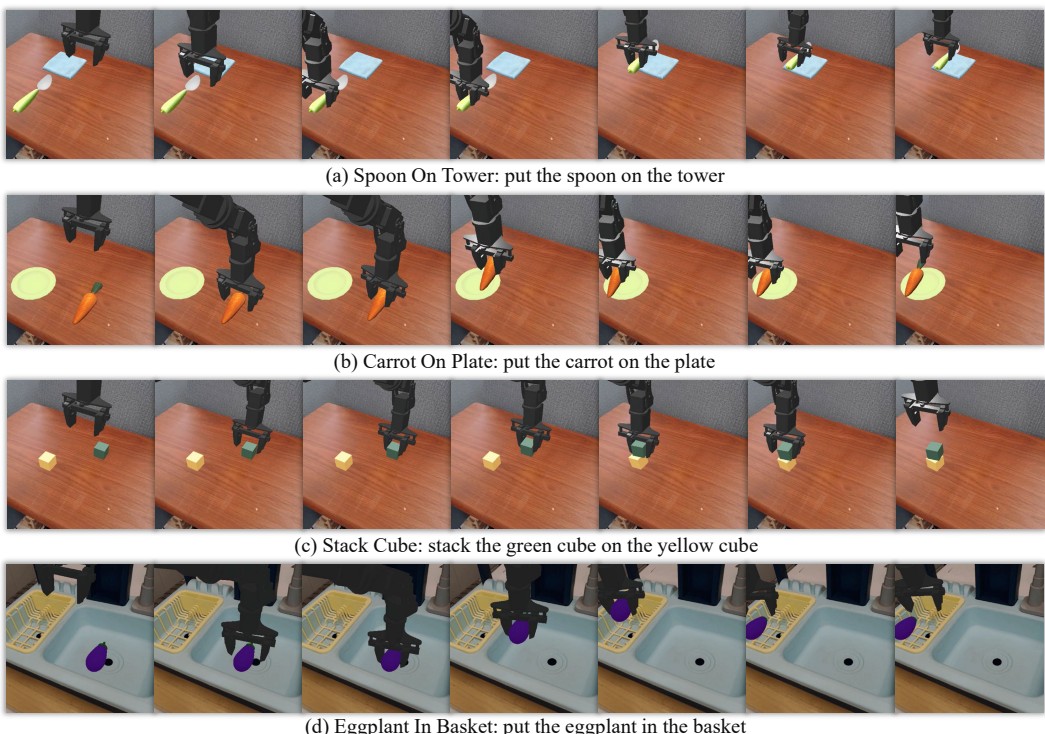

(a) Spoon On Tower: put the spoon on the tower

(b) Carrot On Plate: put the carrot on the plate

(c) Stack Cube: stack the green cube on the yellow cube

(d) Eggplant In Basket: put the eggplant in the basket

Figure 16: **Qualitative results of MemoryVLA on SimplerEnv-Bridge tasks.** Representative examples include *Spoon on Tower*, *Carrot on Plate*, *Stack Cube*, and *Eggplant in Basket* tasks.

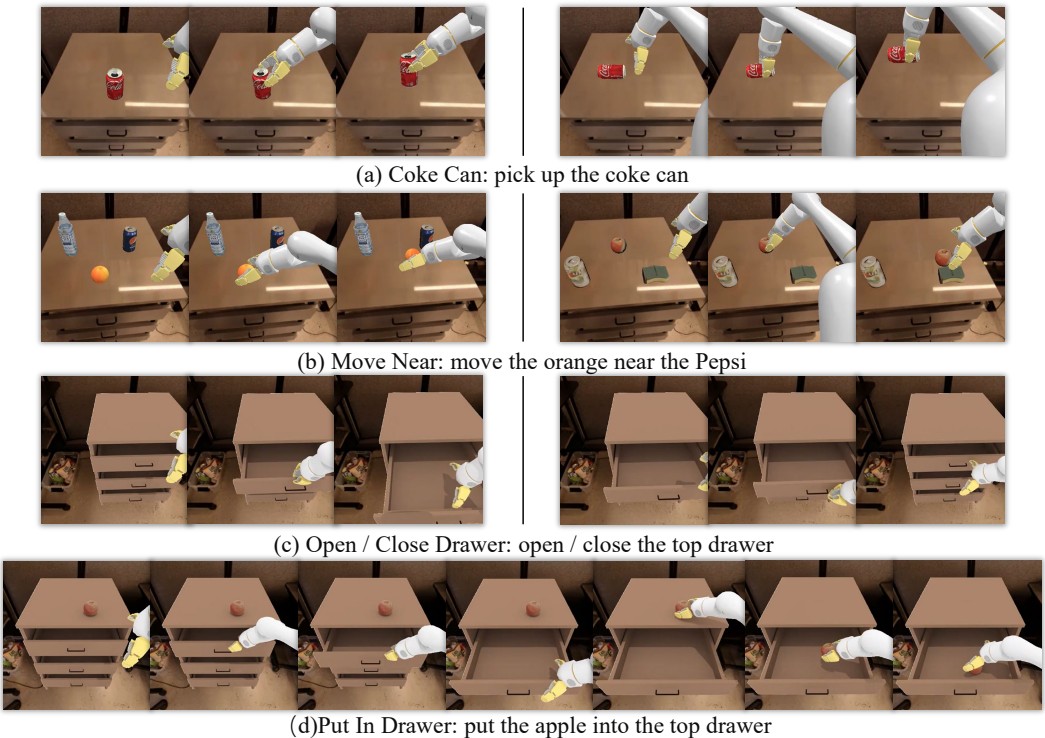

(a) Coke Can: pick up the coke can

(b) Move Near: move the orange near the Pepsi

(c) Open / Close Drawer: open / close the top drawer

(d)Put In Drawer: put the apple into the top drawer

Figure 17: **Qualitative results of MemoryVLA on SimplerEnv-Fractal tasks.** Representative examples include *Pick Coke Can*, *Move Near*, *Open/Close Drawer*, and *Put in Drawer* tasks.

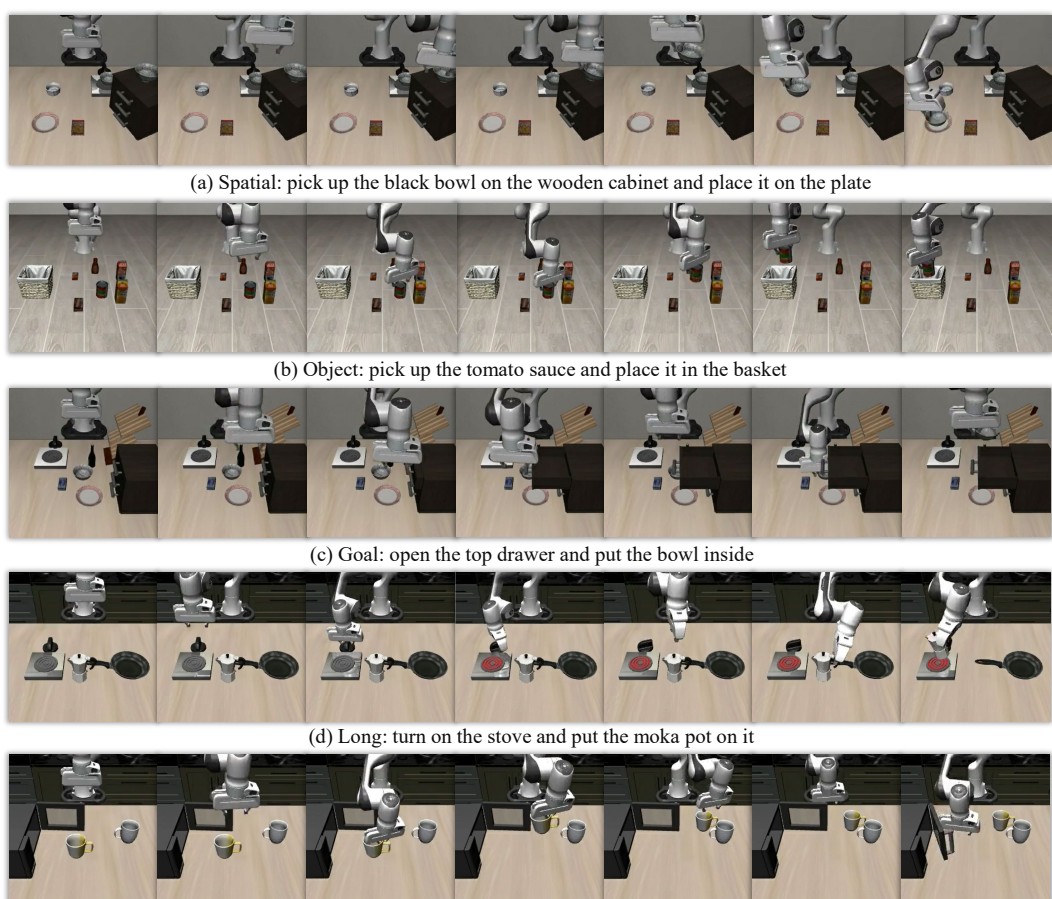

(a) Spatial: pick up the black bowl on the wooden cabinet and place it on the plate

(b) Object: pick up the tomato sauce and place it in the basket

(c) Goal: open the top drawer and put the bowl inside

(d) Long: turn on the stove and put the moka pot on it

(e) LIBERO-90: close the microwave

Figure 18: **Qualitative results of MemoryVLA on LIBERO tasks.** Representative examples include tasks from *Spatial*, *Object*, *Goal*, *Long-10*, and *Long-90* suites.

