# OpenReview forum: "MemoryVLA: Perceptual-Cognitive Memory in Vision-Language-Action Models for Robotic Manipulation"
_ICLR.cc/2026/Conference — ICLR 2026 Poster_

### Official Review · Reviewer_MmT3 · 2025-10-31

**Soundness:** 3
**Presentation:** 3
**Contribution:** 3
**Rating:** 6
**Confidence:** 4

**Summary:**

This paper introduces MemoryVLA, a Cognition-Memory-Action framework for robotic manipulation that addresses the limitation of mainstream VLA models in handling long-horizon, temporally dependent tasks. Inspired by human memory systems, the approach features a Perceptual-Cognitive Memory Bank that consolidates low-level visual details and high-level semantic information, which working memory retrieves and fuses with current observations to condition a diffusion-based action expert.

**Strengths:**

1. The paper identifies a concrete limitation in existing VLA models (ignoring temporal context) and proposes a practical memory bank mechanism with retrieval, fusion, and consolidation operations. The dual-stream design (perceptual + cognitive tokens) with gated fusion is technically sound and the memory consolidation strategy effectively manages computational costs.

2. The method achieves substantial improvements over strong baselines, with comprehensive experiments across 3 robots, and thorough ablations validating each design choice. The real-world deployment on both Franka and WidowX robots demonstrates practical applicability beyond simulation.

**Weaknesses:**

1. The paper does not report inference time and memory footprint comparison with baselines. The memory retrieval via cross-attention at each timestep, especially with memory lengths up to 256, likely incurs significant computational overhead that could limit real-time deployment.

2. The paper lacks visualization or analysis of retrieved memory contents. It's unclear whether the memory bank actually retrieves semantically/temporally relevant contexts or if the gains simply come from having more visual history. Attention weight visualization or case studies showing retrieved frames would strengthen the claims.

3. Ablations (Tables 5-6) are only conducted on SimplerEnv-Bridge. It's unclear if the design choices (e.g., memory length=16, gate fusion, token merge) generalize to other benchmarks like LIBERO or real-world tasks where temporal dependencies may differ.

4. Limited generalization analysis:
* Task generalization: While OOD robustness is tested with visual variations (backgrounds, lighting, occlusion), there's no evaluation of zero-shot generalization to unseen task categories, which is crucial for a general-purpose VLA model.

* Memory capacity analysis: For long-horizon tasks, is memory length of 256 sufficient? The paper doesn't analyze what happens when task horizon exceeds memory capacity, or whether the consolidation strategy causes information loss for very long sequences.

* Benchmark selection for ablations: Tables 5-6 conduct ablations only on SimplerEnv-Bridge, which arguably doesn't require strong temporal dependencies (as the tasks are relatively simple pick-and-place). Ablations should be conducted on benchmarks where temporal reasoning is critical (e.g., real-world long-horizon tasks) to validate design choices.

**Questions:**

See weaknesses.

---

> ### Author Response · Authors · 2025-11-24
> **Respond to Reviewer MmT3 (1 / 2)**
>
> ---
>
> We thank the reviewer for the positive and insightful assessment of our motivation, architecture, and strong experimental results. We appreciate the recognition of our contributions and the detailed feedback.
>
> ---
>
> ### **W1:  Inference Efficiency Comparison**
>
> Thank you for raising the concern about inference time and memory usage. We have added latency, throughput, and GPU memory measurements comparing our method with the baseline on two commonly used GPUs, RTX 4090 and HGX H20. The results below are averaged over 300 runs in bfloat16.
>
> Our method achieves a latency of **0.194** s and a throughput of **82.5 Hz** on RTX 4090, corresponding to only a **~3.6%** increase in overhead compared to the baseline. The memory usage is **16.6 GB**, which is merely **+0.8 GB** over the baseline. These results show that the memory module is lightweight and introduces only negligible extra cost during inference.
>
> The overhead remains small because the memory module is deliberately lightweight: each cognitive memory entry is represented by a single cognitive token, the perceptual memory is compressed to 256 channels through a perceptual compression module, and the memory consolidation step continuously merges redundant entries. Together, these designs keep both the retrieved memory size and cross-attention cost minimal, resulting in negligible additional latency and memory usage during inference.
>
> |   Model   | Latency (RTX4090) | Throughput (RTX4090)  | Latency (HGX H20) | Throughput (HGX H20)  | Memory |
> |:---------:|:-----------------:|:---------------------:|:-----------------:|:---------------------:|:------:|
> | Baseline  |      0.187 s      |       85.6 Hz         |      0.236 s      |       67.8 Hz         | 15.8 GB|
> | MemoryVLA |      0.194 s      |       82.5 Hz         |      0.246 s      |       65.0 Hz         | 16.6 GB|
>
> ---
>
> ### **W2:  Visualization Analysis of Memory Retrieval**
>
> To directly address the reviewer’s request for visualization and analysis of the retrieved memory contents, we added a qualitative study in the **updated Appendix. Fig. 16 (Page 34)** illustrates both the retrieved memory entries and the corresponding attention distributions on representative real-world and simulation tasks.
>
> In the real-world Change Food task, after the first food item is placed aside, the current frame contains two food items on the table, making it impossible to determine from this single observation which one should be picked next. MemoryVLA therefore attends strongly to the nearby frames reflecting the recent motion trend, as well as the last decisive frame before the ambiguity arises.
>
> In the Shell Game Touch task from Mikasa-Robo Simulation Benchmark, the robot is briefly shown the cube location before it is covered by cups. The model consistently attends to the initial revealing frames, which provide the only reliable cue for identifying the correct cup. These results demonstrate that MemoryVLA retrieves meaningful temporal cues essential for disambiguating the next action, rather than simply recalling redundant visual history.
>
> ---
>
> (1 / 2)

---

> ### Author Response · Authors · 2025-11-24
> **Respond to Reviewer MmT3 (2 / 2)**
>
> ---
>
> ### **W3 & W4.3:  Additional Ablation in LIBERO & Real-World Temporal Task**
>
> We thank the reviewer for raising this concern. In the rebuttal, we extended the ablations to both **LIBERO-Long-90** and a **real-world temporal task** (Clean Table & Count). Specifically, we evaluated the effect of memory length, fusion type, consolidation strategy on these benchmarks.
> For memory length, it is closely tied to the episode length of each benchmark. As shown in Tab.15 of the updated appendix, SimplerEnv, LIBERO, and our real-world general tasks have similar episode lengths, whereas real-world temporal tasks are substantially longer. We further conducted an ablation on a representative real-world temporal task (Clean Table \& Count). A moderate memory length (256) performs best, as shown below.
>
> For **memory length**, it is closely tied to the episode length of each benchmark. As shown in Tab.15 of the updated appendix, SimplerEnv, LIBERO, and our real-world general tasks have similar episode lengths, whereas real-world temporal tasks are substantially longer. We further conducted an ablation on a representative real-world temporal task (Clean Table \& Count). A moderate memory length (256) performs best, as shown below.
>
> | Memory Length |  Clean Table & Count (Real-World)  |
> |:-------------:|:----------------------------------:|
> |      64       |                 78                 |
> |  256 (Base)   |               **84**               |
> |     512       |                 81                 |
>
> | Memory Length | LIBERO-Long-90 Tasks |
> |:-------------:|:--------------------:|
> |       8       |        94.2          |
> |   16 (Base)   |      **95.6**        |
> |      32       |        95.6          |
>
> In addition, we extended the ablations on the **memory fusion type** and the **consolidation strategy** to LIBERO-Long-90 and a real-world long-horizon task. The findings are consistent with those observed on the Bridge benchmark.
>
> | Fusion Type | Clean Table & Count (Real-World) | LIBERO-Long-90 Tasks  |
> |:-----------:|:--------------------------------:|:---------------------:|
> |     Add     |               78                 |         93.8          |
> |    Gate     |              **84**              |        **95.6**       |
>
> | Consolidation Type | Clean Table & Count (Real-World) | LIBERO-Long-90 Tasks  |
> |:------------------:|:--------------------------------:|:---------------------:|
> |        FIFO        |               76                 |         94.9          |
> |    Token Merge     |              **84**              |        **95.6**       |
>
> ---
>
> ### **W4.1  Task Generalization**
>
> In addition to visual OOD tests, we have added task generalization experiments to evaluate zero shot performance on unseen task categories. We use Apple To Basket as the base task and test on three unseen tasks: Eggplant To Basket, Blush To Basket, and Apple To Plate. As shown in the table below, our method achieves good zero shot task generalization.
>
> |   Model    | Apple To Basket (Base Task) | Eggplant To Basket (OOD Task) | Blush To Basket (OOD Task) | Apple To Plate (OOD Task) | Avg. Success (OOD Task) |
> |:----------:|:----------------------------:|:-------------------------------:|:----------------------------:|:---------------------------:|:-------------------------:|
> | MemoryVLA  |            4 / 5             |             3 / 5               |            2 / 5             |            2 / 5            |           0.47           |
>
> ---
>
> ### **W4.2  Memory Capacity Analysis**
>
> We analyzed the memory capacity using Clean Table & Count, the longest real-world task with strong temporal dependency (max length=902). The episode length statistics are shown below:
>
> | Task              | Max Length | Median Length | Average Length |
> |:-----------------:|:----------:|:--------------:|:----------------:|
> | Clean Table Count |    902     |      667      |       682       |
>
> We conducted a memory-length ablation on this task and found that **a length of 256 is sufficient**:
>
> | Memory Length |  Clean Table & Count (Real-World)  |
> |:-------------:|:----------------------------------:|
> |      64       |                 78                 |
> |  256 (Base)   |               **84**               |
> |     512       |                 81                 |
>
> This behavior is explained by our Token Merge-based Memory Consolidation strategy. When the memory capacity is exceeded, the two temporally adjacent and most similar memory entries are merged. Because the robot’s motion evolves slowly, consecutive observations are highly redundant. Merging the two features (by summation) preserves their shared content while reducing redundancy. We further computed the cosine similarity of merged feature pairs and found them to **be extremely close (typically ≈0.95)**, confirming that consolidation introduces minimal information loss.
>
> ---
>
> We thank the reviewer again for the thoughtful feedback and would be very happy to continue the discussion.
>
> ---
>
> (2 / 2)

---

> > ### Comment · Reviewer_MmT3 · 2025-11-25
> > **Post-rebuttal comment**
> >
> > The authors addressed my primary concern. I appreciate the authors' response. I will support acceptance during the discussion with AC.

---

> > > ### Author Response · Authors · 2025-11-26
> > >
> > > Thank you for your positive feedback and for supporting the acceptance of our paper. We appreciate your thoughtful comments throughout the review process.

---

### Official Review · Reviewer_5ryq · 2025-10-31

**Soundness:** 2
**Presentation:** 3
**Contribution:** 2
**Rating:** 4
**Confidence:** 4

**Summary:**

The paper tackles core gap in VLA which is weak temporal reasoning for long-horizon, non-markovian manipulation, and they do so by drawing on cognition-memory-action architecture with two complementary memory systems. A pretrained VLM converts observations into (i) perceptual tokens and (ii) higher-level cognitive tokens that together serve as working memory for immediate control. In parallel, a Perceptual-Cognitive Memory Bank accumulates both low-level details and high-level semantic “gist.” At each step, working memory retrieves decision-relevant entries from the bank, adapts/fuses them with current tokens, and updates the bank by merging redundancies. A memory-conditioned diffusion action expert then produces temporally aware action sequences.

**Strengths:**

1. The working-memory vs. long-term (episodic + semantic) split is directly inspired by human memory and mapped cleanly to a VLA stack (perceptual/cognitive tokens + Perceptual-Cognitive Memory Bank). This makes the temporal modeling choice easy to justify and reason about.

2. Converting observations into perceptual and cogntiive tokens enable lightweight retrieval, fusion and consolidations.

3. Good performance on SimplerEnv-Bridge benchmark and LIBERO.

**Weaknesses:**

1,Benchmark mismatch (memory not actually required).

Fundamentally, the simulation benchmark used does not evaluate memory: the tasks appear in-distribution, short-horizon, and solvable without non-Markovian reasoning. I recommend evaluating on a benchmark that explicitly requires memory, such as Memory-Bench (from SAM2Act), to substantiate the paper’s claims.

2.Inadequate baselines (no memory or long-context retrieval).

The chosen baselines are not memory-enhanced and do not leverage long context or retrieval, making it difficult to attribute gains to the proposed method. Please compare against baselines that incorporate memory or retrieval to fairly assess effectiveness.

**Questions:**

The paper should address its two major weaknesses; without doing so, its claims will remain insufficiently supported. I would consider change the rating if my questions can be addressed.

---

> ### Author Response · Authors · 2025-11-24
> **Respond to Reviewer 5ryq (1 / 2)**
>
> ---
>
> We appreciate the reviewer’s recognition of our human-inspired motivation, the overall memory-centric architecture, and the strong performance. We also thank the reviewer for the constructive feedback, including the request to evaluate on explicitly memory-dependent benchmark and to include baselines with temporal context. We provide additional results and analyses accordingly.
>
> ---
>
> ### **W1:  Evaluation on Additional Memory Benchmark**
>
> In addition to the memory-focused real-world tasks already included in the initial submission, we conducted further experiments on the **Mikasa-Robo benchmark**[1] during the rebuttal period.
>
> Mikasa-Robo focuses on memory-dependent tasks and is aligned with the VLA setting, providing VLA-style baselines for fair comparison. MemoryBench, in contrast, primarily uses small policies such as RVT and offers **limited support for VLA-style models**. Moreover, its baselines are already **close to saturation (over 94% success)**, whereas the baselines on Mikasa-Robo remain below 30%. Given this headroom and architectural compatibility, Mikasa-Robo provides a more suitable benchmark for evaluating our method.
>
> In our experiments, we compare MemoryVLA against the VLA-style baselines used in Mikasa-Robo, including OpenVLA-OFT, PI-0, and SpatialVLA. We additionally reproduce CronusVLA, a contemporaneous VLA model that uses temporal context. Following the standard Mikasa-Robo protocol (250 demonstrations per task at 128×128 resolution, end-effector control, and 100 evaluation episodes), MemoryVLA achieves **the highest performance**, with an average improvement of **11.8% over the previous state-of-the-art method**, including **+41.0% on ShellGameTouch task**. The results are shown below.
>
> |     Model     | Shell Game Touch | Intercept Medium | Remember Color3  | Remember Color5  | Remember Color9  | Avg. Success |
> |:-------------:|:----------------:|:----------------:|:----------------:|:----------------:|:----------------:|:------------:|
> |  SpatialVLA   |        23        |        27        |        27        |        17        |        11        |     21.0     |
> |  OpenVLA-OFT  |        47        |        14        |        59        |        16        |         6        |     28.4     |
> |      PI-0     |        33        |        42        |        35        |        22        |        15        |     29.4     |
> |   MemoryVLA   |        88        |        24        |        44        |        30        |        20        |   **41.2**   |
>
> In the initial submission, our real-world evaluation already included several memory-focused, long-horizon tasks (**Tab. 4 and Fig. 10–11**). To make the memory dependency of these tasks clearer, the updated Appendix includes **Fig. 18 and Fig. 19 (Page 38)** which illustrates the memory-relevant aspects of several real-world tasks. Together with the new Mikasa-Robo experiments, these results **across real-world and simulation** demonstrate that MemoryVLA is effective on tasks that rely heavily on memory.
>
> ---
>
> (1 / 2)

---

> ### Author Response · Authors · 2025-11-24
> **Respond to Reviewer 5ryq (2 / 2)**
>
> ---
>
> ### **W2:  More Baseline**
>
> We thank the reviewer for pointing out the need for baselines that explicitly incorporate temporal context.
>
> In the **Mikasa-Robo**[1] benchmark, a memory-dependent manipulation benchmark, no temporal-context VLA baselines are provided in the official release. To ensure a fair comparison, we therefore reproduced **CronusVLA**[2], a contemporaneous strong VLA model that explicitly leverages historical information by aggregating multi-frame VLM features through a sliding-window module. This provides Mikasa-Robo with a temporal-context baseline, and MemoryVLA substantially outperforms it across all tasks.
>
> |   Model    | Shell Game Touch | Intercept Medium | Remember Color3  | Remember Color5  | Remember Color9  | Avg. Success |
> |:----------:|:----------------:|:----------------:|:----------------:|:----------------:|:----------------:|:------------:|
> | CronusVLA  |        32        |         5        | a       31        |        13        |         9        |     18.0     |
> | MemoryVLA  |        88        |        24        |        44        |        30        |        20        |   **41.2**   |
>
> For the **LIBERO** benchmark, we include the main temporal-context VLA baselines commonly used in prior work, including **TTF-VLA**[3], **TraceVLA**[4], **4D-VLA**[5], **CronusVLA**[2], and **MAP-VLA**[6]. These methods represent the strongest publicly available temporal-context VLA models that have reported results on LIBERO. MemoryVLA obtains higher average success rates than these baselines.
>
> |   Method   | Spatial | Object | Goal | Long-10 | Avg. Success |
> |:----------:|:-------:|:------:|:----:|:-------:|:------------:|
> |  TTF-VLA   |  73.0   |  84.0  | 81.0 |  58.0   |     74.0     |
> | TraceVLA   |  84.6   |  85.2  | 75.1 |  54.1   |     74.8     |
> |  4D-VLA    |  88.9   |  95.2  | 90.9 |  79.1   |     88.6     |
> | CronusVLA  |  93.8   |  92.8  | 95.6 |  86.5   |     92.2     |
> |  MAP-VLA   |  96.3   |  98.4  | 95.4 |  83.4   |     93.4     |
> | MemoryVLA  |  98.4   |  98.4  | 96.4 |  93.4   |   **96.7**   |
>
> For the **SimplerEnv** benchmark, we include the temporal-context VLA baselines reported in prior work, including **TraceVLA**[4], **RoboVLMs**[7], and **CronusVLA**[2]. These methods incorporate explicit temporal modeling, and MemoryVLA achieves higher success rates than these baselines.
>
> |   Method   | Spoon on Towel | Carrot on Plate  | Stack Cube  | Eggplant in Basket  | Avg. Success |
> |:----------:|:--------------:|:----------------:|:-----------:|:-------------------:|:------------:|
> | TraceVLA   |      12.5      |       16.6       |    16.6     |        65.0         |     27.7     |
> | RoboVLMs   |      45.8      |       20.8       |     4.2     |        79.2         |     37.5     |
> | CronusVLA  |      66.7      |       54.2       |    20.8     |       100.0         |     60.4     |
> | MemoryVLA  |      75.0      |       75.0       |    37.5     |       100.0         |   **71.9**   |
>
> **Reference**
>
> [1] Cherepanov, E., Kachaev, N., Kovalev, A. K., & Panov, A. I. (2025). Memory, Benchmark & Robots: A Benchmark for Solving Complex Tasks with Reinforcement Learning. arXiv preprint arXiv:2502.10550.
>
> [2] Li, H., Yang, S., Chen, Y., Tian, Y., Yang, X., Chen, X., ... & Pang, J. (2025). CronusVLA: Transferring Latent Motion Across Time for Multi-Frame Prediction in Manipulation. arXiv preprint arXiv:2506.19816.
>
> [3] Liu, C., Zhang, J., Li, C., Zhou, Z., Wu, S., Huang, S., & Duan, H. (2025). Ttf-vla: Temporal token fusion via pixel-attention integration for vision-language-action models. arXiv preprint arXiv:2508.19257.
> [4] Zheng, R., Liang, Y., Huang, S., Gao, J., Daumé III, H., Kolobov, A., ... & Yang, J. (2024). Tracevla: Visual trace prompting enhances spatial-temporal awareness for generalist robotic policies. arXiv preprint arXiv:2412.10345.
>
> [5] Zhang, J., Chen, Y., Xu, Y., Huang, Z., Zhou, Y., Yuan, Y. J., ... & Zhang, L. (2025). 4D-VLA: Spatiotemporal Vision-Language-Action Pretraining with Cross-Scene Calibration. arXiv preprint arXiv:2506.22242.
>
> [6] Li, R., Guo, W., Wu, Z., Wang, C., Deng, H., Weng, Z., ... & Wang, Z. (2025). MAP-VLA: Memory-Augmented Prompting for Vision-Language-Action Model in Robotic Manipulation. arXiv preprint arXiv:2511.09516.
>
> [7] Liu, H., Li, X., Li, P., Liu, M., Wang, D., Liu, J., ... & Zhang, H. (2025). Towards generalist robot policies: What matters in building vision-language-action models.
>
> ---
>
> Overall, we (i) added a memory-focused simulation benchmark and improved by 11.8%, (ii) supplemented each benchmark with strong temporal-context VLA baselines, and (iii) combined these results with our real-robot memory tasks where we observe a +26% gain. These additions directly address the reviewer’s concerns.
>
> We sincerely thank the reviewer for the constructive feedback and are glad to hear the willingness to reconsider the rating. We truly appreciate the comments and look forward to further discussion.
>
> ---
>
> (2 / 2)

---

> > ### Comment · Reviewer_5ryq · 2025-11-26
> > **Rebuttal feedback**
> >
> > Thanks for running the additional experiments, i have a question on if there was an insight why CronusVLA which is memory-centric VLA performed worst than OpenVLA or Pi-0?

---

> > > ### Author Response · Authors · 2025-11-26
> > >
> > > We sincerely thank you for the follow-up question.
> > >
> > > This is indeed a thoughtful observation, and we also noticed this phenomenon. Our main finding is that **CronusVLA struggles to capture the long-term memory dependencies required in Mikasa-Robo**.
> > >
> > > **(1) CronusVLA provides only short-term temporal context.**
> > >
> > > CronusVLA relies on a fixed-size sliding window of past features (e.g., the previous 6 frames) as temporal context. These past features are fused with the current feature and used as the condition for the DiT decoder. However, in robotic manipulation, the motion between consecutive frames is very small, and such a short window does not provide meaningful long-range temporal information. In addition, without a retrieval mechanism, redundant or irrelevant historical features may be mixed into the context and can sometimes negatively affect the decision-making process.
> > >
> > > **(2) Mikasa-Robo tasks depend on long-term memory.**
> > >
> > > For example, as shown in our updated Fig. 16(b) (Page 34), in the ShellGameTouch task from the Mikasa-Robo Benchmark, the correct decision depends on observations made much earlier in the episode. Since CronusVLA only retains a small number of recent frames, the crucial information has already slid out of the window, making it unable to capture such long-term dependencies. In practice, it mostly captures short-term temporal cues.
> > >
> > > **(3) A minor factor is architectural differences.**
> > >
> > > CronusVLA and our MemoryVLA both use a LLaMA-2 + DiT architecture, while PI-0 adopts a flow-matching framework and OpenVLA-OFT uses a purely autoregressive model. These architectural choices and their corresponding pretraining pipelines may introduce some differences in performance.
> > >
> > > We appreciate the reviewer bringing up this question, and we hope the above analysis helps clarify the result.

---

### Official Review · Reviewer_PZt6 · 2025-11-01

**Soundness:** 1
**Presentation:** 3
**Contribution:** 2
**Rating:** 2
**Confidence:** 5

**Summary:**

The paper introduces a new model, MemoryVLA, which employs a specialized memory bank designed to better handle temporal dependencies.

**Strengths:**

1. Incorporating memory mechanisms into VLAs is a highly relevant and important research direction.
2. The paper presents a large number of experiments, including those conducted on a real robot.
3. The work is well written and easy to read.

**Weaknesses:**

1. Despite the large number of experiments, the main drawback of the paper is that most of the tasks used do not actually require a memory mechanism.  The authors should conduct comparisons on specialized robotics benchmarks focused on memory-based tasks, such as Mikasa-Robo [1] and MemoryBench [2]. Without these experiments, it is impossible to properly evaluate the effectiveness of the proposed memory mechanism.
2. The results on LIBERO outperform Discrete Diffusion VLA [3] by only 0.3, even though the latter does not use any memory mechanisms. This again raises questions about the suitability of the chosen benchmarks for evaluation.
3. The functioning of the memory mechanism is not demonstrated clearly, it can only be inferred indirectly from the overall model performance. It is important to show that the memory bank retrieves relevant elements, providing direct evidence of how the mechanism contributes to task solving.

In its current state, I believe that, despite addressing an important direction, the paper does not sufficiently demonstrate that the proposed memory mechanism truly helps solve complex memory-dependent tasks. This is primarily due to testing mostly on simple tasks that do not require memory, as well as the lack of an in-depth analysis of the memory mechanism itself.

I am willing to reconsider my evaluation if these shortcomings are addressed.

References:
1. Cherepanov, Egor, et al. "Memory, Benchmark & Robots: A Benchmark for Solving Complex Tasks with Reinforcement Learning." arXiv preprint arXiv:2502.10550 (2025).
2. Fang, Haoquan, et al. "Sam2act: Integrating visual foundation model with a memory architecture for robotic manipulation." arXiv preprint arXiv:2501.18564 (2025).
3. Liang, Zhixuan, et al. "Discrete diffusion vla: Bringing discrete diffusion to action decoding in vision-language-action policies." arXiv preprint arXiv:2508.20072 (2025).

**Questions:**

1. The main question is how the proposed model performs on tasks from Mikasa-Robo [1] and MemoryBench [2].
2. In [1], it was shown that using an action chunk of large size can circumvent the need for a memory mechanism to solve tasks. In MemoryVLA, a chunk of size T = 16 is used, predicting 16 steps ahead. How would the model behave if this value were reduced? In which cases does performance improvement come from using a long chunk, and in which cases from the memory mechanism itself?
3. What is the number of steps (actions) required to solve the tasks (mean, minimum, maximum, and median), including on the real robot?
4. How do the elements retrieved from the memory bank correspond to the task being solved at a given moment?
5. Why does the proposed model perform so much better than CogACT in real-robot experiments compared to simulation? Were the models trained under comparable conditions?

---

> ### Author Response · Authors · 2025-11-24
> **Respond to Reviewer PZt6 (1 / 3)**
>
> ---
>
> We thank the reviewer for the detailed feedback and for recognizing the importance of memory mechanisms in VLA models. Some concerns appear to come from focusing mainly on the simulation benchmarks, whereas our real-world evaluation includes several explicitly memory-dependent temporal tasks. We address each point below with additional clarifications and evidence.
>
> ---
>
> ### **W1 & Q1:  Evaluation on Additional Memory Benchmark**
>
> We thank the reviewer for the suggestion to evaluate on simulation benchmark explicitly designed for memory-dependent tasks. Beyond our real-world memory-focused tasks in the initial submission, we conducted **further experiments on the Mikasa-Robo benchmark** during the rebuttal period. Mikasa-Robo Benchmark focuses on memory-dependent tasks and is aligned with the VLA setting, making it suitable for evaluating our method.
>
> In our experiments, we compare MemoryVLA with VLA-style baselines used in Mikasa-Robo, including OpenVLA-OFT, PI-0, and SpatialVLA. Following the standard Mikasa-Robo protocol using 250 demonstrations per task at 128×128 resolution, 100 evaluation episodes per task and end-effector control. MemoryVLA consistently achieves **the highest performance**, with an average of **11.8% improvement over the previous state-of-the-art method**, especially **+41.0% on ShellGameTouch task**. The results are shown below.
>
> |    Model     | Shell Game Touch | Intercept Medium | Remember Color3  | Remember Color5  | Remember Color9  | Avg. Success  |
> |:------------:|:----------------:|:----------------:|:----------------:|:----------------:|:----------------:|:-------------:|
> |  SpatialVLA  |        23        |        27        |        27        |        17        |        11        |     21.0      |
> | OpenVLA-OFT  |        47        |        14        |        59        |        16        |         6        |     28.4      |
> |     PI-0     |        33        |        42        |        35        |        22        |        15        |     29.4      |
> |  MemoryVLA   |        88        |        24        |        44        |        30        |        20        |   **41.2**    |
>
> ---
>
> ### **W2:  LIBERO Results and Benchmark Suitability**
>
> The small margin over Discrete Diffusion VLA on LIBERO is expected. First, Discrete Diffusion VLA uses richer visual inputs than our model, taking **both a wrist view and a third-person view**, while MemoryVLA uses **only a single third-person view**. Second, most LIBERO tasks now achieve 90%+ success rate with recent VLA models, indicating that the benchmark is **close to saturation** and naturally leaves limited room for further improvement.
>
> We report LIBERO because it remains one of the mainstream benchmarks for evaluating **general manipulation**. Across these general tasks, MemoryVLA consistently maintains strong performance, and the added temporal consistency from the memory mechanism still yields moderate gains (for example, **+14.6%** on SimplerEnv-Bridge and **+9%** on real-world general tasks).
>
> For **evaluating tasks that rely heavily on memory**, we include **both real-world and simulation** results. In the real world, our long-horizon, memory-dependent tasks show a **+26% improvement (Tab. 4 and Fig. 10–11)**. In simulation, MemoryVLA obtains clear gains on **Mikasa-Robo** Benchmark (**+11.8%** over the previous SOTA method), and we also report results on LIBERO-90, which contains numerous long-horizon tasks. Together, these results demonstrate that MemoryVLA provides clear benefits on tasks where memory is essential.
>
> ---
>
> (1 / 3)

---

> ### Author Response · Authors · 2025-11-24
> **Respond to Reviewer PZt6 (2 / 3)**
>
> ---
>
> ### **W3 & Q4:  Evidence of How the Memory Mechanism Functions**
>
> To provide a direct view of how the memory mechanism functions, the updated Appendix includes a qualitative case study. **Fig. 16 (Page 34)** visualizes the **retrieved memory elements and their attention scores** on the real-world and simulation tasks. The model consistently attends to past frames that resolve decision-relevant ambiguities absent from the current observation.
>
> In the **real-world** Change Food task, after the first food item is placed aside, the current frame contains two food items on the table, making it impossible to determine from this single observation which one should be picked next. MemoryVLA therefore attends strongly to the nearby frames reflecting the recent motion trend, as well as the last decisive frame before the ambiguity arises.
>
> In the Shell Game Touch task from **Mikasa-Robo simulation benchmark**, the robot is briefly shown the cube location before it is covered by cups. The model consistently attends to the initial revealing frames, which provide the only reliable cue for identifying the correct cup. These results demonstrate that MemoryVLA retrieves meaningful temporal cues essential for disambiguating the next action, rather than simply recalling redundant visual history.
>
> ---
>
> ### **Q2:  Action Chunking vs. Memory Mechanism**
>
> We thank the reviewer for raising this very relevant question. Action chunking and the memory mechanism play fundamentally different roles. Action chunking mainly improves **short-horizon temporal smoothness**, while the memory module models **long-horizon temporal dependencies** that cannot be inferred from the current observation. In other words, action chunking cannot substitute for memory.
>
> To verify this, we conducted an ablation reducing the action chunk length from 16 to 4 on Mikasa-Robo benchmark. The overall performance shows **only a very small drop**. Only the highly dynamic InterceptMedium task exhibits a more visible decrease, where short-horizon trajectory smoothness naturally matters more. Overall, reducing the chunk length has minimal impact on performance, indicating that the improvements of MemoryVLA **do not come from long action chunks**.
>
> In contrast, OpenVLA-OFT shows **a large drop** when the chunk length is reduced, because chunking is its only source of temporal information in **the absence of a memory mechanism**.
>
> |           Model            | Shell Game Touch | Intercept Medium  | Remember Color3  | Remember Color5  | Remember Color9  | Avg. Success  |
> |:--------------------------:|:----------------:|:-----------------:|:----------------:|:----------------:|:----------------:|:-------------:|
> | OpenVLA-OFT (Chunk Len=4)  |        12        |         6         |        21        |         9        |         8        |      9.3      |
> | OpenVLA-OFT (Chunk Len=8)  |        47        |        14         |        59        |        16        |         6        |     28.4      |
> | MemoryVLA (Chunk Len=4)    |        84        |        17         |        47        |        30        |        20        |     39.6      |
> | MemoryVLA (Chunk Len=16)   |        88        |        24         |        44        |        30        |        20        |    **41.2**   |
>
> ---
>
> ### **Q3:  Action Length Statistics**
>
> The table below reports the maximum, minimum, median, and average action lengths across all task suites, including SimplerEnv Evaluation (Bridge, Fractal), LIBERO (Spatial/Object/Goal and 10/90 task suites), and both real-world general and temporal tasks. For the real-world tasks, we additionally provide filtered statistics based on a motion-magnitude threshold (translation>1 cm or rotation>0.4 rad between consecutive frames) to remove frames where the end-effector motion is small.
>
> |        Task Suite           |   Max   |  Min   | Median | Average |
> |:---------------------------:|:-------:|:------:|:------:|:-------:|
> | SimplerEnv-Bridge / Fractal |   200   |   80   |  117   |   119   |
> | LIBERO-Spatial / Object / Goal | 270  |   75   |  131   |   130   |
> |        LIBERO-10 / 90       |   505   |   58   |  144   |   156   |
> |   Real-General (Original)   |  1575   |  281   |  575   |   575   |
> |   Real-General (Filtered)   |   213   |   40   |   81   |    84   |
> |  Real-Temporal (Original)   |  7704   |  412   |  981   |  1672   |
> |  Real-Temporal (Filtered)   |   902   |   72   |  236   |   288   |
>
> ---
>
> (2 / 3)

---

> ### Author Response · Authors · 2025-11-24
> **Respond to Reviewer PZt6 (3 / 3)**
>
> ---
>
> ### **Q5:  Real-Robot Performance**
>
> **Training conditions are fair**. Both models use the same RealSense D435 third-person RGB input with identical 224×224 preprocessing, no wrist images or robot states. The training data, number of evaluation rollouts, training steps, learning rate, augmentation pipeline (Appendix D.4), data sampling strategy, action chunking, and all key hyperparameters are exactly matched, and both models follow the same evaluation protocol. The performance gap therefore does not arise from training differences.
>
> The larger improvement on real-robot temporal tasks reflects that these tasks **contain stronger memory dependencies**. When the next action cannot be inferred from the current observation alone, the benefit of memory becomes much more pronounced. On real-world general tasks, the gain is modest (+9%). On memory-focused simulation tasks such as Mikasa-Robo, MemoryVLA shows clear improvements (+11.8%). The largest gain appears on real-robot long-horizon temporal tasks (+26%), which aligns with their stronger reliance on temporal memory.
>
> To clarify **why real-world temporal tasks require memory**, in the updated Appendix, **Fig.18 (Page 38)** highlights the key moments in several tasks where the correct action depends on past information rather than the current observation. **Fig.19 (Page 38)** further provides a step-by-step example from the Change Food task, showing a case where the next action becomes ambiguous from a single frame and can only be resolved by recalling earlier steps. These examples illustrate that **the stronger gains on real-robot temporal tasks stem from their inherently memory-dependent nature**.
>
> ---
>
> ### **Concluding Remarks**
>
> We again thank the reviewer for the detailed feedback and for recognizing the importance of memory mechanisms in VLA models. Some of the concerns seem to arise from how the memory demands of our tasks were interpreted, and we hope the additional analyses and experiments help clarify this.
>
> For tasks that **rely strongly on memory**, we evaluate in **both simulation and the real world**.
> - MemoryVLA improves over the previous state of the art by **11.8% on the Mikasa-Robo benchmark**,
> - and by **26% on the real-world** long-horizon temporal tasks,
> directly demonstrating the effectiveness of the memory mechanism.
>
> For **general manipulation tasks**, we evaluate on SimplerEnv-Bridge, Fractal, LIBERO, and real-world general tasks. MemoryVLA provides **consistent performance gains** in these settings as well, showing that memory modeling also enhances temporal consistency for general tasks.
>
> We also include **several analyses that make the role of memory explicit**:
> - a case study about how the memory mechanism functions,
> - an ablation about action chunking and memory,
> - trajectory-length statistics across all benchmarks,
> - and visual explanations of the memory dependencies in our real-robot tasks.
>
> Together, these results show that MemoryVLA benefits both memory-dependent and general tasks, and they directly address the questions raised in the review.
>
> We appreciate the reviewer’s openness to reconsidering the evaluation and hope that the additional experiments and analyses are helpful.
>
> ---
>
> (3 / 3)

---

### Official Review · Reviewer_F2kP · 2025-11-01

**Soundness:** 3
**Presentation:** 4
**Contribution:** 3
**Rating:** 6
**Confidence:** 4

**Summary:**

The paper tackles non-Markovian, long-horizon robotic manipulation where single-frame VLAs fail. It proposes MemoryVLA, a Cognition-Memory–Action framework that (i) converts a single RGB frame + instruction into perceptual tokens (DINOv2+SigLIP compressed via SE-bottleneck) and a single cognitive token (EOS from LLaMA-7B), (ii) stores both streams in a Perceptual–Cognitive Memory Bank (PCMB), and (iii) performs retrieval (cross-attention + timestep PE), gate fusion, and consolidation (adjacent-pair merge by cosine similarity) before a DiT+DDIM action head predicts a 16-step continuous 7-DoF trajectory. The authors conduct an extensive evaluation across 150+ tasks in simulation (SimplerEnv, LIBERO) and the real world.

**Strengths:**

* Clear structure with good motivation: the task of handling of non-Markovian tasks is interesting and significant in robotics, where the motivation fused memory term within the architecture design.
* Extensive evaluation: the authors evaluate MemoryVLA across three different robots, three distinct simulation benchmarks (SimplerEnv-Bridge, SimplerEnv-Fractal, LIBERO), and a set of 12 real-world tasks. This comprehensive evaluation on 150+ tasks with 500+ variations provides high confidence in the method's effectiveness and generalizability.
* Significant performance gain: the model achieves state-of-the-art performance across all benchmarks. The standout result is the +26 point improvement over the next-best baseline on real-world long-horizon tasks.

**Weaknesses:**

* The ambiguity of optimal memory length: from the ablation study in Table 5, it suggests that a memory length of $L=16$ is optimal (71.9% success), while the performance worsens at $L=64$ (67.7%). However, in the Appendix, the authors state that a memory length of $L=256$ was used for real-world long-horizon tasks. There lacks of in-depth analysis of how memory length is associated with the actual performance.
* The mechanism of using single cognitive token: for complex tasks require multiple latent hypotheses, one EOS token could be lossy. The paper doesn’t probe whether more cognitive capacity helps or hurts.

**Questions:**

* Although the model shows strong robustness to many OOD variations, the performance drops sharply when viewpoint changes in Appendix (Sec. C). Does that suggest the learned perceptual features are highly view-dependent and that the memory module may be memorizing visual details rather than an abstracted spatial representation?
* How does the memory length affect the performance of the model?
* How does different memory modules work independently? Can you provide a qualitative example of a task failure that occurs with only cognitive memory (i.e., lacking perceptual detail) and a failure that occurs with only perceptual memory (i.e., lacking cognitive gist)?

---

> ### Author Response · Authors · 2025-11-24
> **Respond to Reviewer F2kP**
>
> ---
>
> We sincerely appreciate the reviewer’s thoughtful assessment and the recognition of our motivation, architectural design, and strong performance across the diverse simulation and real-world evaluations. We address the questions and comments below.
>
> ---
>
> ### **W1 & Q2:  Impact of Memory Length on Model Performance**
>
> Thank you for raising this important point. The choice of memory length is closely tied to the **episode length of each benchmark**. As shown in the table below, SimplerEnv, LIBERO, and our real-world general tasks have similar episode lengths, whereas real-world temporal tasks are substantially longer, even after heavy frame filtering. This naturally motivates using a larger memory length for temporal tasks.
>
> | Task Suite                      | Max Length | Average Length |
> |:-------------------------------:|:----------:|:--------------:|
> | SimplerEnv-Bridge / Fractal     |    200     |      119       |
> | LIBERO-Spatial / Object / Goal  |    270     |      130       |
> | LIBERO-10 / 90                  |    505     |      156       |
> | Real-General (Original)         |    1575    |      575       |
> | Real-General (Filtered)         |    213     |       84       |
> | Real-Temporal (Original)        |    7704    |      1672      |
> | Real-Temporal (Filtered)        |    902     |      288       |
>
> We further conducted an ablation on a representative **real-world temporal task** (Clean Table & Count). A moderate memory length (256) performs best.
>
> | Memory Length |  Clean Table & Count (Real-World)  |
> |:-------------:|:----------------------------------:|
> |      64       |                 78                 |
> |  256 (Base)   |               **84**               |
> |     512       |                 81                 |
>
> We also evaluated memory length on **LIBERO-Long-90 tasks**. Even with a larger memory length, the improvement remains marginal, and for consistency across all LIBERO suites, we keep the memory length fixed at 16.
>
> | Memory Length | LIBERO-Long-90 Tasks |
> |:-------------:|:--------------------:|
> |       8       |        94.2          |
> |   16 (Base)   |      **95.6**        |
> |      32       |        95.6          |
>
> ---
>
> ### **W2:  Number of Cognitive Tokens**
>
> We added an ablation on the number of cognitive tokens. As shown in the table below, increasing the count from 1 to 4 brings no performance gain. Perhaps the single 4096-dim EOS token already captures sufficient semantic information, so adding more tokens does not provide additional benefit.
>
> | Num. Cog Token | Spoon on Towel | Carrot on Plate  | Stack Cube | Eggplant in Basket  | Avg. Success  |
> |:--------------:|:--------------:|:----------------:|:----------:|:-------------------:|:-------------:|
> |   1 (Base)     |      75.0      |       75.0       |    37.5    |       100.0         |    **71.9**   |
> |       4        |      79.2      |       66.7       |    37.5    |        95.8         |     69.8      |
>
> ---
>
> ### **Q1:  Camera View Generalization**
>
> The drop under camera-view changes is not caused by the memory module. **View generalization is primarily determined by the visual encoder and the diversity of training viewpoints, not by the memory mechanism.** Our model is trained with **a single fixed third-person RGB view**, so the learned perceptual features naturally become view-dependent, as also observed in prior works.
> This effect is independent of the memory bank.
>
> In our early real-world experiments, we found that models trained with data from multiple camera viewpoints naturally acquire strong cross-view generalization. Since prior VLA baselines adopt a fixed-camera training setup, we follow the same setting in this work.
>
> ---
>
> ### **Q3:  Failure Cases with Single Memory**
>
> Perceptual memory captures local past details, while cognitive memory stores global, high-level semantics. As shown in the **updated Appendix (Fig. 17, Page 37)**, the Clean Table & Count task clearly illustrates their independent failure modes. In this task, the robot removes each item and must press the button once per removal to record the count.
>
> **Failure with Only Cognitive Memory.**
> After clearing an item, the robot moves its arm downward to press the button. Without perceptual memory to retain the local motion history, the model cannot distinguish whether the arm is in the process of moving down to press, or has already pressed and is now moving upward. As a result, it may lift too early and never press the button, causing a missed count.
>
> **Failure with Only Perceptual Memory.**
> With only perceptual memory, the robot can press the button properly and complete the counting action. However, without cognitive memory, it does not know whether it has already pressed the button. As a result, it may press multiple times for a single cleared item, or miss the press for an item, leading to double-counting or skipped counts.
>
> ---
>
> We thank the reviewer again for the thoughtful feedback and look forward to further discussion.
>
> ---

---

### Author Response · Authors · 2025-11-24
**Global Response: Paper Revisions**

We thank all reviewers for the valuable comments. According to the reviewers’ suggestions, all newly added analyses and experiments are included in **Appendix J (Page. 32–38)**. Our method proposes a memory architecture for VLA inspired by human hippocampal memory mechanisms, and the newly added results further validate its effectiveness across both memory-focused and general manipulation tasks.

- In Sec. J.1, we added a new Mikasa-Robo memory-focused benchmark with an average improvement of +11.8%.
- In Sec. J.2, we added temporal-context VLA baselines on Mikasa, LIBERO, and Simpler.
- In Sec. J.3, we added a case study visualizing memory retrieval.
- In Sec. J.4, we added an analysis of action chunking and memory mechanisms.
- In Sec. J.5, we reported data length statistics.
- In Sec. J.6, we added extensive ablations on memory length, fusion type, consolidation strategy, and cognitive token count across LIBERO-Long-90 and real-world memory tasks.
- In Sec. J.7, we added inference efficiency results.
- In Sec. J.8, we added zero-shot task generalization.
- In Sec. J.9, we analyzed failure cases with single memory.
- In Sec. J.10, we explained the memory dependence of real-world tasks.

Across benchmarks, MemoryVLA achieves **+11.8% on simulation memory tasks, +26% on real-world memory tasks**, +14.6% on the general Bridge benchmark, and +9% on general real-world tasks.

We have also provided detailed point-by-point responses to each reviewer. We sincerely thank the reviewers for their detailed feedback and hope the revisions address the concerns raised.

---

### Author Response · Authors · 2025-12-04
**To AC: Rebuttal Summary**

---
Dear AC, SAC, PC,

Thank you very much for your time and for supporting the review process. We appreciate the reviewers’ constructive feedback. During the rebuttal, we addressed **all concerns** and updated the manuscript accordingly, with new results placed in **Appendix J (Pages 32–38)**.

The reviewers recognized the **value** of introducing memory into VLA, the insight of **motivation** and **architectural design**, and the **strong performance** across both simulation and real-robot. Their main concerns were:
- Lack of a memory-focused simulation benchmark
  - We added **Mikasa-Robo in simulation**, where MemoryVLA improves over prior SOTA by **11.8%**, complementing the existing memory-focused **real-robot** results (**+26%**).
- Need for additional ablations, visualizations, baselines, and analyses
  - We added all corresponding materials in Appendix J.
Below is a concise summary of our responses.

---
### Reviewer F2kP (Initial Score: 6)
- Concern: Missing ablation studies
  - We added several new ablations covering memory length and cognition token number. (Tab. 15-17, Page 35)
- Concern: Camera view generalization
  - We clarified the issue stems from the use of a single fixed camera view. View generalization is governed by the visual encoder and the diversity of training viewpoints, not by the memory mechanism itself.
- Concern: Missing failure cases with single memory
  - We added detailed visualizations. (Fig. 17, Page 37)

---
### Reviewer PZt6 (Inital Score: 2)
- Concern: Lack of memory-focused simulation benchmark
  - Our original submission already included **memory-focused real-robot evaluations**.
  - In the rebuttal, we additionally introduced a **simulation benchmark, Mikasa-Robo**, where our method outperforms the prior SOTA by **+11.8**. (Tab. 12, Page 32)
- Concern: Benchmark suitability
  - After the rebuttal, our evaluation is comprehensive across **both general and memory-focused benchmarks, in both simulation and real-world**. Our method achieves substantial gains:
    - General: Simulation (Bridge) **+14.6**, Real robot **+9**
    - Memory-focused: Simulation (Mikasa-Robo): **+11.8**, Real robot: **+26**
- Concern: Missing visualization of the memory retrieval mechanism
  - We added a visualization in Fig. 16 (Page 34), showing that our memory module consistently attends to the key task-relevant information.
- Concern: Missing analysis on chunk size and memory
  - On Mikasa-Robo, models with memory stay stable under smaller chunk sizes, while models without memory drop sharply. This shows the gains come from memory, not action chunking. (Tab. 14, Page 34)
  - Memory captures long-range dependencies; chunking only smooths short-term actions.
- Concern: Action length statistics
  - We provided the action length statistics. (Tab. 15, Page 35).
- Concern: LIBERO and real-robot performance
  - LIBERO is nearly saturated (95+), so the gains are naturally smaller. Since we use only the front-view (no wrist or state inputs), the improvement margin is further limited.
  - Memory-focused real-robot tasks benefit much more due to their strong reliance on memory (Fig. 18-19, Page 38).

We appreciate the reviewer indicated they would **reconsider their score**.

---
### Reviewer 5ryq (Inital Score: 4)
- Concern: Lack of memory-focused simulation benchmark
  - We add the **Mikasa-Robo benchmark**, where our method outperforms the prior SOTA by **+11.8** (Tab. 12, Page 32). Together with the memory-focused real-robot results already in the original submission, this further verifies the method’s effectiveness.
- Concern: Lack of temporal-context baselines
  - We add **CronusVLA, TTF-VLA, TraceVLA, 4D-VLA, MAP-VLA, and RoboVLMs** across Mikasa-Robo, Bridge, and LIBERO (Tab. 13, Page 33).

We also appreciate the reviewer’s openness to **revising their score**.

---
### Reviewer MmT3 (Inital Score: 6)
- Concern: Inference Efficiency
  - We added a comparison showing that the memory mechanism adds only ~3% inference time and memory overhead. (Tab. 20, Page 36)
- Concern: Visualization of memory retrieval
  - We added it (Fig. 16, Page 34).
- Concern: Additional ablations
  - We added ablations on memory length, memory fusion, and the consolidation (Tab. 15, 16, 18, 19; Pages 35–36).
- Concern: Task generalization
  - We added real-world task generalization results. (Tab. 21, Page 36).
- Concern: Memory capacity analysis
  - We clarified the memory consolidation mechanism effectively handles over-capacity cases, with supporting ablations in Tab. 16 (Page 35).

We appreciate the reviewer confirmed their concerns were **fully addressed and expressed support for acceptance**.

---
We sincerely appreciate everyone involved in the review process, especially the AC, for their efforts in this very unusual year. We deeply appreciate the time and dedication invested in evaluating our work.

---

### Meta-Review · Area_Chair_jjeR · 2026-01-15

**Summary:**

Summary: The initial reviews were split. Reviewers F2kP and MmT3 recognized the novelty of the Cognition-Memory-Action architecture and the strong performance on general benchmarks (SimplerEnv, LIBERO), rating the paper as a weak accept. However, Reviewers PZt6 and 5ryq raised significant concerns regarding the suitability of the evaluation benchmarks. They argued that the chosen tasks were largely Markovian and solvable without memory, requesting comparisons on explicit memory-dependent benchmarks (specifically Mikasa-Robo) and stronger comparisons against temporal/memory-based baselines. There were also concerns regarding inference latency, the interpretability of the retrieval mechanism, and the necessity of the single cognitive token.

**Reviewer Concerns:**

Addressed: The authors provided a highly responsive rebuttal.

- Benchmark Suitability (PZt6, 5ryq): The authors conducted new experiments on the requested Mikasa-Robo benchmark, demonstrating SOTA performance (+11.8% over baselines) and resolving the concern that the method was not tested on true memory tasks.
- Baselines (5ryq, PZt6): The authors added comparisons to temporal-context VLAs (CronusVLA, TraceVLA, TTF-VLA, etc.) across simulation and real-world tasks, showing consistent superiority.
- Mechanism Interpretability (PZt6, MmT3): The added visualizations of memory attention weights (e.g., attending to past frames to resolve occlusion in Shell Game) effectively demonstrated the module's function.
- Inference Cost (MmT3): The provided latency/memory analysis (only ~3.6% overhead) addressed efficiency concerns.
- Ablations (F2kP): The authors justified the cognitive token count and memory length through extensive ablations.

Outstanding:

- View Dependence (F2kP): The authors acknowledged that view generalization is limited by the fixed-camera training data rather than the memory architecture. While explained, the limitation remains inherent to the current implementation.

**Reviewer Scores:**

- Reviewer F2kP (Score: 6 -> 6). The authors addressed the specific questions regarding memory length ambiguity and cognitive token capacity. The rebuttal strengthened the technical soundness of the paper.

- Reviewer PZt6 (Score: 2 -> 4 or 6). This reviewer explicitly stated they were willing to reconsider if the authors evaluated on Mikasa-Robo or MemoryBench. The authors executed this exact request on Mikasa-Robo with strong results. The "Reject" was predicated on the lack of memory-specific evidence, which is now present.

- Reviewer 5ryq (Score: 4 -> 6). This reviewer engaged in the discussion regarding CronusVLA's performance. The authors provided a sound technical explanation regarding context window length vs. long-term dependencies. As the reviewer stated they would change their rating if questions were addressed, and the baseline/benchmark issues were resolved, a move to Accept is logical.

- Reviewer MmT3 (Score: 6 -> 8). The reviewer explicitly commented post-rebuttal that their primary concerns were addressed and they "will support acceptance".

---

### Decision · Program_Chairs · 2026-01-26

Accept (Poster)